# PnP Inversion: Boosting Diffusion-based Editing with 3 Lines of Code

**Xuan Ju**[1,2*], **Ailing Zeng**[2†], **Yuxuan Bian**[1], **Shaoteng Liu**[1], **Qiang Xu**[1†]

[1]The Chinese University of Hong Kong (CUHK) [2]International Digital Economy Academy (IDEA)

{xju22,stliu21,qxu}@cse.cuhk.edu.hk {zengailing}@idea.edu.cn

*A white horse running in the field* → *Watercolor of a white horse running in the field*

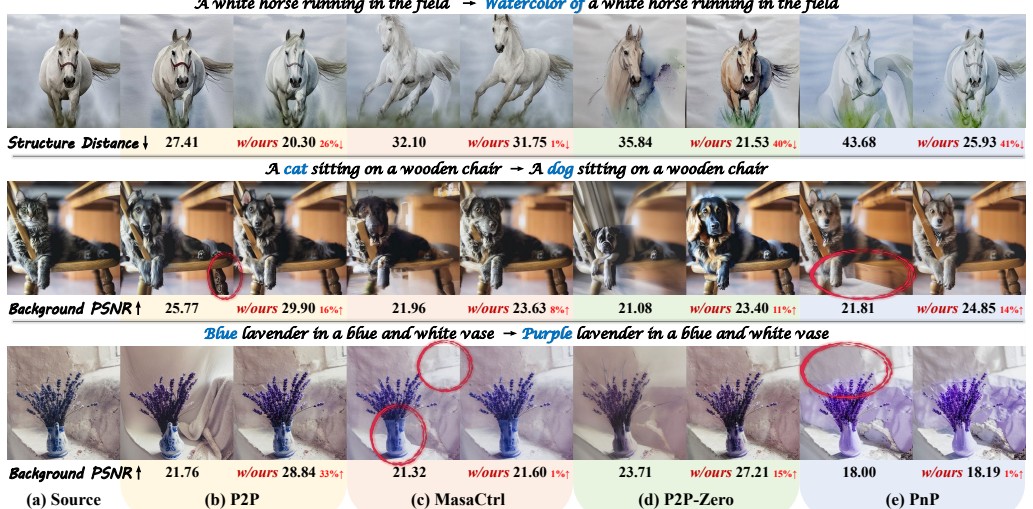

| Structure Distance ↓ | 27.41 | *w/ours* 20.30 26%↓ | 32.10 | *w/ours* 31.75 1%↓ | 35.84 | *w/ours* 21.53 40%↓ | 43.68 | *w/ours* 25.93 41%↓ |

*A cat sitting on a wooden chair* → *A dog sitting on a wooden chair*

| Background PSNR ↑ | 25.77 | *w/ours* 29.90 16%↑ | 21.96 | *w/ours* 23.63 8%↑ | 21.08 | *w/ours* 23.40 11%↑ | 21.81 | *w/ours* 24.85 14%↑ |

*Blue lavender in a blue and white vase* → *Purple lavender in a blue and white vase*

| Background PSNR ↑ | 21.76 | *w/ours* 28.84 33%↑ | 21.32 | *w/ours* 21.60 1%↑ | 23.71 | *w/ours* 27.21 15%↑ | 18.00 | *w/ours* 18.19 1%↑ |
| (a) Source | (b) P2P | (c) MasaCtrl | (d) P2P-Zero | (e) PnP |

Figure 1: **Performance enhancement of incorporating *PnP Inversion* into four diffusion-based editing methods** across various editing categories (from top to bottom): style transfer, object replacement, and color change. The editing prompt is displayed at the top of each row, which includes (a) the source image, the editing results of (b) Prompt-to-Prompt (P2P) (Hertz et al., 2023), (c) MasaCtrl (Cao et al., 2023), (d) pix2pix-zero (Parmar et al., 2023), and (e) plug-and-play (Tumanyan et al., 2023). Each set of results is presented: the first column w/o *PnP Inversion* (Null-text inversion for P2P, DDIM Inversion for the others), and the second column w/ *PnP Inversion*. Incorporating *PnP Inversion* into diffusion-based editing methods results in improved image structure preservation (enhancement of the structure distance metric) for full image editing and enhanced background preservation (increased PSNR metric values in the background, i.e., areas that should remain unedited) for foreground editing. The improvements are mostly tangible, and we circle some of the subtle discrepancies w/o *PnP Inversion* in red. **Best viewed with zoom in.**

## Abstract

Text-guided diffusion models have revolutionized image generation and editing, offering exceptional realism and diversity. Specifically, in the context of diffusion-based editing, where a source image is edited according to a target prompt, the process commences by acquiring a noisy latent vector corresponding to the source image via the diffusion model. This vector is subsequently fed into separate source and target diffusion branches for editing. The accuracy of this inversion process significantly impacts the final editing outcome, influencing both *essential content preservation* of the source image and *edit fidelity* according to the target prompt.

Prior inversion techniques aimed at finding a unified solution in both the source and target diffusion branches. However, our theoretical and empirical analyses reveal that disentangling these branches leads to a distinct separation of responsibilities for preserving essential content and ensuring edit fidelity. Building on this insight, we introduce "*PnP Inversion*," a novel technique achieving optimal performance of both branches with just three lines of code. To assess image editing performance, we present *PIE-Bench*, an editing benchmark with 700 images showcasing diverse scenes and editing types, accompanied by versatile annotations and comprehensive evaluation metrics. Compared to state-of-the-art optimization-based inversion techniques, our solution not only yields superior performance across 8 editing methods but also achieves nearly an order of speed-up.

---

*This work was done when Xuan Ju was intern at IDEA.

†Corresponding author.

# 1 INTRODUCTION

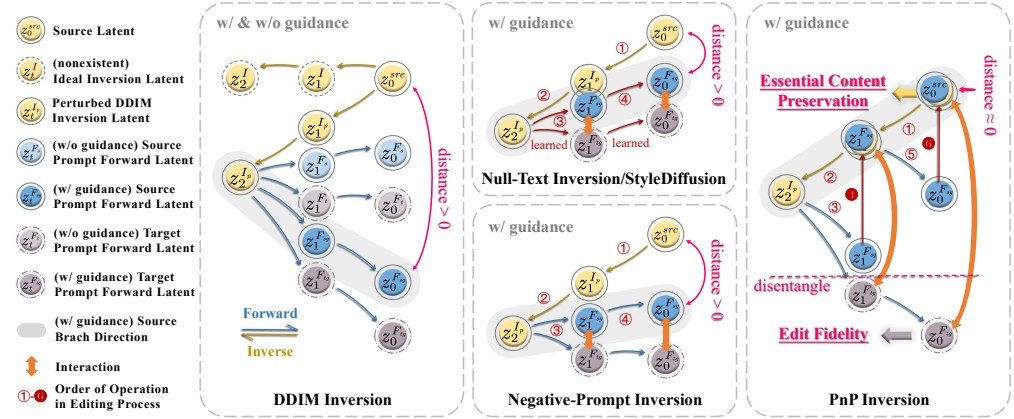

Figure 2: **Comparisons among different inversion methods in diffusion-based editing.** We assume a 2-step diffusion process for illustration. Due to nonexistent of ideal $z_2^I$, common practice uses DDIM Inversion (Song et al., 2020) to approximate $z_t^I$, resulting in $z_t^{I_p}$ with perturbation. Diffusion-based editing methods start from the perturbed noisy latent $z_2^{I_p}$ and perform DDIM sampling in a source and a target diffusion branch, further resulting in the distance shown on the figure. Null-Text Inversion and StyleDiffusion optimize a specific latent used in both source and target branches to reduce this distance. Negative-Prompt Inversion assigns the guidance scale to 1 to decrease the distance. In contrast, *PnP Inversion* disentangles source and target branches in editing. By leaving the target diffusion branch untouched, *PnP Inversion* retains the edit fidelity. By directly returning the source branch to $z_0^{src}$, *PnP Inversion* achieves the best possible essential content preservation. We use numbers to mark operation step order, where solid circles are steps added by *PnP Inversion*.

Text-guided diffusion models (Rombach et al., 2022; Ramesh et al., 2022) have become the mainstream image generation technique, praised for their realism and diversity. As the noise latent space of diffusion models (Meng et al., 2022; Kawar et al., 2023; Hertz et al., 2023; Balaji et al., 2022; Liu et al., 2023a) possesses the capacity to retain and modify images, we can perform prompt-based editing with diffusion models, where a source image is edited according to a target prompt. The common practice is to maintain two diffusion branches: one for the source image and the other for the target image. By carefully exchanging information between the two branches, we can preserve the essential content in the source image while achieving edit fidelity according to the target prompt. However, such manipulations are only straightforward when the diffusion latent space (noisy latent in each diffusion step) corresponding to the source image is available. When editing images without known latent space, we have to invert the diffusion model to obtain their latent vectors first.

While DDIM inversion is effective for unconditional diffusion (Song et al., 2020; Dhariwal & Nichol, 2021), much of the research (Hertz et al., 2023; Han et al., 2023) has centered on inverting the diffusion process with conditional inputs. This is driven by the significance of conditions in applications like text-based image editing. However, introducing conditions undermines DDIM inversion quality, as evidenced in Figure 2. With the advent of Null-Text Inversion (Mokady et al., 2023), a prevailing consensus (Dong et al., 2023; Li et al., 2023b) has emerged: achieving superior inversion[1] necessitates rigorous optimization. Methods that forgo such optimization, such as Negative-Prompt Inversion (Miyake et al., 2023), compromise editing outcomes. In this paper, we challenge this prevailing wisdom, posing two fundamental questions: **What exactly are these optimization-based inversion methods truly aiming at? And, are such optimizations indispensable for diffusion-based image editing?**

As illustrated in Figure 2, prior optimization-based approaches strive to minimize the distance between $z_0^{src}$ and $z_0^{F_{sg}}/z_0^{F_{tg}}$ by indirectly tweaking the generation model's input parameters. Given the magnitude of the optimization network, like UNet, and the impracticality of prolonged optimization durations, these methods often optimize the target latent with just a few iterations. This results

---

[1] A more suitable name for them would be inversion correction techniques. But we follow previous works and call them inversion techniques.

in a learned latent with a discernible gap between $z_0^{F_{sg}}/z_0^{F_{tg}}$ and the original $z_0^{src}$. Moreover, the learned latent does not appear during the generation models' training process, deviating from the pretrained diffusion model's original input distribution. Such forced input assignments hinder the model's generative capacity, compromising the integrity of both the source and editing branches.

In this work, we delve into the intricacies of text-based inversion, providing a thorough analysis of existing techniques. Our theoretical and empirical analyses demonstrate that the exhaustive optimization in prior techniques is, counterintuitively, not only far from optimal but unnecessary. Introducing *PnP Inversion* (**P**lug-a**n**d-**P**lay **Inversion**), our approach offers a simple yet potent inversion solution for diffusion-based editing. The essence of *PnP Inversion* lies in two primary strategies: (1) **disentangle** the source and target branches, and (2) empower each branch to **excel in its designated role**: preservation or editing. Specifically, the source branch in *PnP Inversion* rectifies the deviation path directly, using only 3 lines of code. This addresses the challenges seen in earlier approaches: (1) undesirable latent space distances affecting **essential content preservation**, (2) misalignment in the generation model's distribution, and (3) extended processing times. For the target branch, we keep it unchanged, ensuring the best possible **edit fidelity** in line with the target prompt.

Addressing the lack of standardized benchmarks for inversion and editing, we introduce **PIE-Bench** (**P**rompt-based **I**mage **E**diting **Bench**mark) with 700 images from diverse scenes, spanning 10 editing categories. Each entry consists of a source prompt, target prompt, editing directive, edit subjects, and a hand-annotated editing mask. To assess *PnP Inversion* and benchmark it against existing techniques, we employ 7 metrics emphasizing both essential content preservation and edit fidelity. Compared with 5 inversion methods with Prompt-to-Prompt editing, *PnP Inversion* outperforms them, enhancing essential content preservation by up to $83.2\%$ in Structure Distance and edit fidelity by up to $8.8\%$ in Edit Region Clip Similarity, while achieving nearly an order of editing speedup over optimization-based inversion methods. Moreover, across 8 editing approaches, *PnP Inversion* boosts content preservation by as much as $20.2\%$ and edit fidelity by up to $2.5\%$ relative to their baseline configurations. Visualization results are shown in Figure 1.[2]

## 2 RELATED WORK

Diffusion-based image editing aims to manipulate images with diffusion models using given instructions such as natural language descriptions (Hertz et al., 2023), point dragging (Shi et al., 2023), and semantic masking (Meng et al., 2022). This involves two primary concerns: (1) edit fidelity: ensure the editing aligns with the provided instructions; and (2) essential content preservation: inverse the images, particularly the regions that do not require modification, into diffusion latent space while ensuring accurate reconstruction during the editing procedure. Details are in supplementary files.

For edit fidelity, previous methods perform editing roughly through four ways: (1) end-to-end editing model (Brooks et al., 2023; Kim et al., 2022; Nichol et al., 2022; Geng et al., 2023) that trains end-to-end diffusion models to edit images , (2) latent integration (Meng et al., 2022; Avrahami et al., 2022; 2023; Couairon et al., 2023; Zhang et al., 2023b; Shi et al., 2023; Joseph et al., 2023) that inserts editing instruction through the noisy latent feature in target diffusion branch to source diffusion branch. , (3) attention integration (Hertz et al., 2023; Han et al., 2023; Parmar et al., 2023; Cao et al., 2023; Tumanyan et al., 2023; Zhang et al., 2023a; Mou et al., 2023) that fuses attention map connecting the text and image in the source and editing diffusion branch, and (4) target embedding (Kawar et al., 2023; Cheng et al., 2023; Wu et al., 2023; Brack et al., 2023; Tsaban & Passos, 2023; Valevski et al., 2022; Dong et al., 2023; Wu & De la Torre, 2022; 2023) that aggregates editing information of the target branch into an embedding and then insert it to source diffusion branch.

For essential content preservation, previous methods tried to retain the source image's feature through (1) overfit the editing image (Kawar et al., 2023; Shi et al., 2023) so that editing will not make massive changes to the image content, (2) DDPM/DDIM inversion variation (Miyake et al., 2023; Huberman-Spiegelglas et al., 2023; Wallace et al., 2023) to strengthen the source image's influence on both the source and target branch by modifying DDPM/DDIM inversion formulation, (3) attention preservation (Mou et al., 2023; Cheng et al., 2023; Cao et al., 2023; Parmar et al., 2023; Tumanyan et al., 2023; Hertz et al., 2023; Qi et al., 2023) that retains the attention map feature of the source diffusion branch during attention map fusion of the source and target branches, and (4) source embedding (Mokady et al., 2023; Dong et al., 2023; Li et al., 2023b; Gal et al., 2022; Fei et al., 2023; Huang et al., 2023) that absorbs the background or image information to an embedding and use this embedding to reconstruct essential content of the source image.

---

[2]Code is available at `https://github.com/cure-lab/PnPInversion`.

## 3 PRELIMINARIES

This section will first introduce DDIM sampling and classifier-free guidance commonly employed in Diffusion Models. Then, we will delve into the issues arising from the utilization of classifier-free guidance and DDIM sampling in editing, and show how previous works address these challenges.

**Diffusion Models.** Diffusion models aim to map a random noise vector $z_T$ to a series of noise samples $z_t$ and, finally, an output image or latent $z_0$ by adding Gaussian noise $\epsilon$ step-by-step, where $t \sim [1, T]$ and $T$ is the timestep number. To train diffusion models, we first sample $z_t$ from a real image following equation 1 where $\epsilon \sim \mathcal{N}(0, 1)$ and $\alpha$ is hyper-parameter.

$$z_t = \sqrt{\alpha_t} z_0 + \sqrt{1 - \alpha_t} \epsilon \tag{1}$$

Then, a denoiser network $\epsilon_\theta$ is optimized to predict $\epsilon(z_t, t)$ with the objective:

$$\min_\theta E_{z_0, \epsilon \sim \mathcal{N}(0, I), t \sim Uniform(1, T)} \| \epsilon - \epsilon_\theta(z_t, t) \| \tag{2}$$

To generate images from given $z_T$, we employ the deterministic DDIM sampling (Song et al., 2020):

$$z_{t-1} = \frac{\sqrt{\alpha_{t-1}}}{\sqrt{\alpha_t}} z_t + \sqrt{\alpha_{t-1}} \left( \sqrt{\frac{1}{\alpha_{t-1}} - 1} - \sqrt{\frac{1}{\alpha_t} - 1} \right) \epsilon_\theta(z_t, t) \tag{3}$$

**DDIM Inversion.** Although diffusion models have superior characteristics in the feature space (Balaji et al., 2022; Dong et al., 2023; Feng et al., 2022) that can support various downstream tasks, similar to GAN-based models (Xia et al., 2022), it is hard to apply them to images in the absence of natural diffusion feature space for non-generated images. Thus, a technique inverting $z_0^{src}$ back to $z_T$ (i.e., $z_T^I$) is necessary. To address this, a straightforward inversion technique known as DDIM inversion is commonly used for unconditional diffusion models, predicated on the presumption that the ODE process can be reversed in the limit of infinitesimally small steps.

$$z_t^I = \frac{\sqrt{\alpha_t}}{\sqrt{\alpha_{t-1}}} z_{t-1}^I + \sqrt{\alpha_t} \left( \sqrt{\frac{1}{\alpha_t} - 1} - \sqrt{\frac{1}{\alpha_{t-1}} - 1} \right) \epsilon_\theta(z_{t-1}^I, t - 1) \tag{4}$$

However, in most diffusion models, this presumption cannot be guaranteed, resulting in a perturbation from $z_t^I$ to $z_t^{I_p}$ in equation 3, equation 4 and Figure 2. Consequently, an additional perturbation from $z_t^{I_p}$ to $z_t^{F_s}/z_t^{F_t}$ arises when utilizing equation 1 to sample an image from $z_T^{I_p}$ shown in Figure 2.

**Classifier-free Guidance.** To insert conditions, Ho et al. (Ho & Salimans, 2022) present classifier-free guidance, which predicts noise both unconditionally and conditionally, then mix them together:

$$\epsilon_\theta(z_t, t, C, \oslash) = w \cdot \epsilon_\theta(z_t, t, C) + (1 - w) \cdot \epsilon_\theta(z_t, t, \oslash), \tag{5}$$

where $w$ is the guidance scale, $C$ is the condition (embedding of text prompt in our task), and $\oslash$ is the null condition (embedding of null in our task). This further leads to another perturbation from $z_t^{F_s}/z_t^{F_t}$ to $z_t^{F_{sg}}/z_t^{F_{tg}}$ due to the destruction of the DDIM process as demonstrated in Figure 2.

**Previous Inversion Techniques.** Currently, the predominant inversion technique employed for reducing the adverse impact caused by DDIM inversion and classifier-free guidance is optimization-based methods, such as Null-Text Inversion (Mokady et al., 2023) and StyleDiffusion (Li et al., 2023b). Alternative inversion techniques, such as Edit Friendly DDPM (Huberman-Spiegelglas et al., 2023), Negative-Prompt Inversion (Miyake et al., 2023), and EDICT (Wallace et al., 2023), exhibit instability in editing outcomes in both essential content preservation and edit fidelity. The qualitative and quantitative results in our experiment further corroborate this instability.

Optimization-based inversion methods use a specific latent variable to reduce the difference between $z_t^{F_{sg}}$ and $z_t^{I_p}$. For example, Null-Text Inversion revises equation 5 to equation 6 and learns the specific latent variable by gradient propagation using the loss function $z_t^{F_{sg}} - z_t^{I_p}$. This learned variable will be further used in both the source and target branches in editing.

$$\epsilon_\theta(z_t, t, C, \oslash) = w \cdot \epsilon_\theta(z_t, t, C) + (1 - w) \cdot \epsilon_\theta(z_t, t, variable) \tag{6}$$

# 4 METHOD

## 4.1 MOTIVATION

We explain our motivation by raising and then answering two questions.

**Why do optimization-based methods perform better among previous inversion methods?**

Edit Friendly DDPM (Huberman-Spiegelglas et al., 2023) proposes an alternative latent noise space by changing the DDPM sampling distribution to help reconstruction of the desired image. Negative-Prompt Inversion (Miyake et al., 2023) assigns conditioned text embedding to Null-Text embedding and thus maintains a guidance scale of 1 to reduce the deviation in editing. EDICT (Wallace et al., 2023) maintains two coupled noise vectors to invert each other in an alternating fashion for image reconstruction, which reduces the editability of diffusion models. Compared with these inversion techniques, optimization-based inversion (Mokady et al., 2023; Li et al., 2023b; Dong et al., 2023) does not influence the distribution in DDIM sampling (compared to Edit Friendly DDPM), retains enough guidance for text conditions (compared to Negative-Prompt Inversion), and maintains the diffusion model's editability (compared to EDICT).

**Are such optimizations indispensable and optimal for diffusion-based image editing?**

Optimization-based inversion methods learn a specific latent variable to minimize the loss function $z_t^{I_p} - z_t^{F_{sg}}$. Thus, the target of optimization-based inversion is to correct $z_t^{F_{sg}}$ back to $z_t^{I_p}$. The learned latent variable then serves as an input for both the source and target branches.

The optimization of a unified variable for source and target branches leads to several problems: (1) To optimize the specific latent variable, a prolonged processing time is needed during inference (*e.g.*, 148.48 seconds per image for Null-Text Inversion), which is impractical for editing tasks with user interaction; (2) Considering that extended optimization times are not expected, previous approaches have opted to optimize the target latent for only a limited number of iterations. Consequently, the result frequently entails a learned latent space with a discernible gap between $z_0^{F_{sg}}$ and the initial $z_0^{src}$, especially when a large distance exists between $z_t^{F_{sg}}$ and $z_t^{I_p}$. This leads to a decline in essential content preservation ability, as shown in our ablation study; (3) The learned variable serves as the generation model's input parameter, which is not aligned with the diffusion model's expected input distribution and leads to negative impacts on the diffusion model integrity and $z_t^{F_{tg}}$. These three issues hinder the practicality and editability of these optimization-based inversion methods.

## 4.2 METHOD

Bearing these issues into consideration, we propose *PnP Inversion*. The **key** of *PnP Inversion* is to **disentangle the source and target branch**, thus enabling each branch to unleash its maximum potential individually. In the source branch, we can directly add $z_t^{I_p} - z_t^{F_{sg}}$ back to $z_t^{F_{sg}}$, which is a simple strategy that can directly rectify the deviation path and is plug-and-play to various editing methods. In the target branch, simply leaving it unaltered would maximize the diffusion models' potential for target image generation. This simple but effective solution solves the three issues in optimization-based inversion by (1) No optimization is required, thus incurring minimal additional time overhead; (2) Adding $z_t^{I_p} - z_t^{F_{sg}}$ eliminates the discernible gap between $z_0^{F_{sg}}$ and the initial $z_0^{src}$; (3) Do not have any impact on the distribution of the diffusion model's input and $z_t^{F_{tg}}$.

Algorithm 1 presents the algorithm for inserting *PnP Inversion* into existing diffusion-based image editing methods. Red lines with gray backgrounds highlight the 3 lines of code added by *PnP Inversion*. Typical diffusion-based editing of images involves two parts: an inversion process to get the diffusion space of the image, and a forward process to perform editing on the diffusion space.

Specifically, given a source image or latent embedding $z_0^{src}$, we first use DDIM Inversion with source prompt $C^{src}$ to obtain the perturbed diffusion space $z_t^{I_p}$ in Part I (Algorithm 1 line 1-5). Then, in Part II (Algorithm 1 line 6-8), existing editing methods take in source prompt $C^{src}$ and target prompt $C^{tgt}$ simultaneously as input with a batch size of 2 to perform essential content preservation and editing by integrating information between two batches. We use [] to enclose the elements in a batch for better illustration. E.g., [a,b] represents a batch with a and b concatenate in batch dimension.

*PnP Inversion* can be plug-and-played into the forward process in Part II and rectifies the deviation path step by step as shown in Algorithm 1 line 9-13. Particularly, line 10 computes the perturbed

forward latent $z_{t-1}^{F_{sg}}$ of the source branch in DDIM forward as shown in Figure 2. Then, $d_t^{rec}$, the difference between the perturbed DDIM Inversion latent $z_{t-1}^{I_p}$ and $z_{t-1}^{F_{sg}}$, the DDIM forward latent with $C^{src}$ as conditions, is calculated in line 11. Finally, in line 12, we add the $d_t^{rec}$ to and only to the source/reconstruction branch in $\text{DDIM\_Forward}_{\text{Editing\_Model}}$, which is the key to rectifying the deviation path as well as retaining the editability of the target prompt's latent space.

---

**Algorithm 1:** Real Image Editing Pipeline with *PnP Inversion*

---

**Input:** A source prompt embedding $C^{src}$ (or embedding of null for some editing methods), a target prompt embedding $C^{tgt}$, a real image or latent embedding $z_0^{src}$, a prompt-based image editing method $\text{DDIM\_Forward}_{\text{Editing\_Model}}$ (e.g., Prompt-to-Prompt)

**Output:** A reconstruction image or latent embedding $z_0^{F_{sg}}$, an edited image or latent embedding $z_0^{F_{tg}}$

---

**Part I : Invert $z_0^{src}$**

1   $z_0^{I_p} = z_0^{src}$;

2   **for** $t = 1, \ldots, T-1, T$ **do**

3     $\left[ z_t^{I_p} \right] \leftarrow \text{DDIM\_Inversion}\left( \left[ z_{t-1}^{I_p} \right], t-1, \left[ C^{src} \right] \right)$;

4   **end**

5   $\left[ z_T^{F_{sg}}, z_T^{F_{tg}} \right] = \left[ z_T^{I_p}, z_T^{I_p} \right]$;

---

**Part II: Perform editing on $z_T^{I_p}$**

6   **for** $t = T, T-1, \ldots, 1$ **do**

7     $\left[ z_{t-1}^{F_{sg}}, z_{t-1}^{F_{tg}} \right] \leftarrow \text{DDIM\_Forward}_{\text{Editing\_Model}}\left( \left[ z_t^{rec}, z_t^{tgt} \right], t, \left[ C^{src}, C^{tgt} \right] \right)$

8   **end**

---

**Part II: Perform editing on $z_T^{I_p}$ with *PnP Inversion***

9   **for** $t = T, T-1, \ldots, 1$ **do**

10     $\left[ \boldsymbol{z_{t-1}^{F_{sg}}} \right] \leftarrow \text{DDIM\_Forward}\left( \left[ \boldsymbol{z_t^{I_p}} \right], \boldsymbol{t}, \left[ \boldsymbol{C^{src}} \right] \right)$ ;        // 1

11     $\left[ \boldsymbol{d_{t-1}^{rec}} \right] \leftarrow \left[ \boldsymbol{z_{t-1}^{I_p}} \right] - \left[ \boldsymbol{z_{t-1}^{F_{sg}}} \right]$ ;        // 2

12     $\left[ z_{t-1}^{F_{sg}}, z_{t-1}^{F_{tg}} \right] \leftarrow \text{DDIM\_Forward}_{\text{Editing\_Model}}\left( \left[ z_t^{rec}, z_t^{tgt} \right], t, \left[ C^{src}, C^{tgt} \right] \right) \boldsymbol{+ [d_{t-1}^{rec}, 0]}$ ;    // 3

13   **end**

---

14   **Return** $z_0^{F_{sg}}, z_0^{F_{tg}}$

---

### 4.3   BENCHMARK CONSTRUCTION

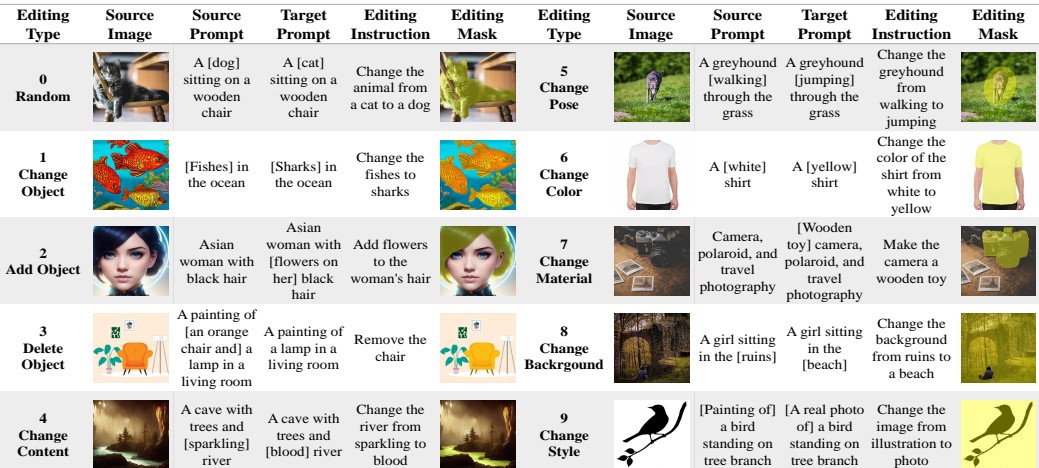

**Figure 3: Examples of *PIE-Bench*.** One example is provided for each editing type. For each sample, we have provided corresponding annotations of editing type, source image, source prompt, target prompt, editing instruction, and editing mask (the mask region is in yellow).

While diffusion-based editing has garnered significant attention in recent years, evaluations of various editing methods have primarily relied on subjective and limited visualizations. To systematically validate our proposed method as a plug-and-play strategy for editing models and compare our method with existing inversion methods, as well as compensate for the absence of standardized

performance criteria for inversion and editing techniques, we construct a benchmark dataset, named *PIE-Bench* (**P**rompt-based **I**mage **E**diting **Bench**mark).

*PIE-Bench* comprises 700 images featuring 10 distinct editing types. Images are evenly distributed in natural and artificial scenes (*e.g.*, paintings) among four categories: animal, human, indoor, and outdoor. Each image in *PIE-Bench* includes five annotations: source image prompt, target image prompt, editing instruction, main editing body, and the editing mask. Notably, the editing mask annotation (indicating the anticipated editing region) is crucial in accurate metrics computations as we expect the editing to only occur within a designated area. Details are in the supplementary files.

## 5 EXPERIMENTS

Due to page limitation, we only provide the comparison of inversion-based editing, essential content preservation methods, ablation on *PnP Inversion* and Null-Text Inversion, and the influence of adding the difference to target latent in this section. More experiments are in supplementary files.

### 5.1 EVALUATION METRICS

To illustrate the effectiveness and efficiency of our proposed *PnP Inversion*, we use eight metrics covering four aspects: structure distance (Tumanyan et al., 2022), background preservation (PSNR, LPIPS (Zhang et al., 2018), MSE, and SSIM (Wang et al., 2004) outside the annotated editing mask), edit prompt-image consistency (CLIPSIM (Wu et al., 2021) of the whole image and regions in the editing mask) and inference time. Details can be found in the supplementary files.

### 5.2 COMPARISON WITH INVERSION-BASED EDITING

| Method | | Structure | | Background Preservation | | | | CLIP Similariy | |
| :---: | :---: | :---: | :---: | :---: | :---: | :---: | :---: | :---: | :---: |
| **Inverse** | **Editing** | **Distance**$_{\times 10^3}$ ↓ | **PSNR** ↑ | **LPIPS**$_{\times 10^3}$ ↓ | **MSE**$_{\times 10^4}$ ↓ | **SSIM**$_{\times 10^2}$ ↑ | | **Whole** ↑ | **Edited** ↑ |
| **DDIM** | **P2P** | 69.43 | 17.87 | 208.80 | 219.88 | 71.14 | | 25.01 | **22.44** |
| **NT**[†] | **P2P** | 13.44 | 27.03 | 60.67 | 35.86 | 84.11 | | 24.75 | 21.86 |
| **NP** | **P2P** | 16.17 | 26.21 | 69.01 | 39.73 | 83.40 | | 24.61 | 21.87 |
| **StyleD** | **P2P** | **11.65** | 26.05 | 66.10 | 38.63 | 83.42 | | 24.78 | 21.72 |
| **Ours** | **P2P** | **11.65**$_{83\%↓}$ | **27.22**$_{52\%↑}$ | **54.55**$_{74\%↓}$ | **32.86**$_{85\%↓}$ | **84.76**$_{19\%↑}$ | | **25.02**$_{1.7\%↑}$ | 22.10$_{1.7\%↑}$ |
| **DDIM** | **MasaCtrl** | 28.38 | 22.17 | 106.62 | 86.97 | 79.67 | | 23.96 | 21.16 |
| **Ours** | **MasaCtrl** | **24.70**$_{13\%↓}$ | **22.64**$_{2\%↑}$ | **87.94**$_{18\%↓}$ | **81.09**$_{7\%↓}$ | **81.33**$_{2\%↑}$ | | **24.38**$_{1.8\%↑}$ | **21.35**$_{0.9\%↑}$ |
| **DDIM** | **P2P-Zero** | 61.68 | 20.44 | 172.22 | 144.12 | 74.67 | | 22.80 | 20.54 |
| **Ours** | **P2P-Zero** | **49.22**$_{20\%↓}$ | **21.53**$_{5\%↑}$ | **138.98**$_{19\%↓}$ | **127.32**$_{12\%↓}$ | **77.05**$_{3\%↑}$ | | **23.31**$_{2.2\%↑}$ | **21.05**$_{2.5\%↑}$ |
| **DDIM** | **PnP**[*] | 28.22 | 22.28 | 113.46 | 83.64 | 79.05 | | **25.41** | 22.55 |
| **Ours** | **PnP**[*] | **24.29**$_{14\%↓}$ | **22.46**$_{1\%↑}$ | **106.06**$_{7\%↓}$ | **80.45**$_{4\%↓}$ | **79.68**$_{1\%↑}$ | | **25.41** | **22.62**$_{0.3\%↑}$ |

[*] use Stable Diffusion v1.5 as base model (others all use Stable Diffusion v1.4)
[†] averaged results on A800 and RTX3090 since different environment leads to slightly different performance

Table 1: **Compare *PnP Inversion* with other inversion techniques across various editing methods.** For editing method Prompt-to-Prompt (P2P) (Hertz et al., 2023), we compare four different inversion methods: DDIM Inversion (DDIM) (Song et al., 2020), Null-Text Inversion (NT) (Mokady et al., 2023), Negative-Prompt Inversion (NP) (Miyake et al., 2023), and StyleDiffusion (StyleD) (Li et al., 2023b). For editing methods MasaCtrl (Cao et al., 2023), Pix2Pix-Zero (P2P-Zero) (Cao et al., 2023), Plug-and-Play (PnP) (Tumanyan et al., 2023), we compare with DDIM Inversion (DDIM).

This section compares *PnP Inversion* with previous inversion-based editing methods quantitatively and qualitatively. Four inversion methods, DDIM Inversion (Song et al., 2020), Null-Text Inversion (Mokady et al., 2023), Negative-Prompt Inversion (Miyake et al., 2023), and StyleDiffusion (Li et al., 2023b), as well as four editing methods, Prompt-to-Prompt (Hertz et al., 2023), MasaCtrl (Cao et al., 2023), pix2pix-zero (Parmar et al., 2023), and Plug-and-Play (Tumanyan et al., 2023) are taken into consideration. For inversion methods such as EDICT (Wallace et al., 2023) and Edit-Friendly DDPM Inversion (Huberman-Spiegelglas et al., 2023), we put them into section 5.3 since their main target is to preserve the background and result in a decline in editability.

| Method | Time (s) |
| :---: | :---: |
| NP | **18.22** |
| EF | 19.10 |
| AIDI | 35.41 |
| EDICT | 35.48 |
| NT | 148.48 |
| StyleD | 382.98 |
| Ours | 28.17 |

Table 2: **Comparison of inference time.**

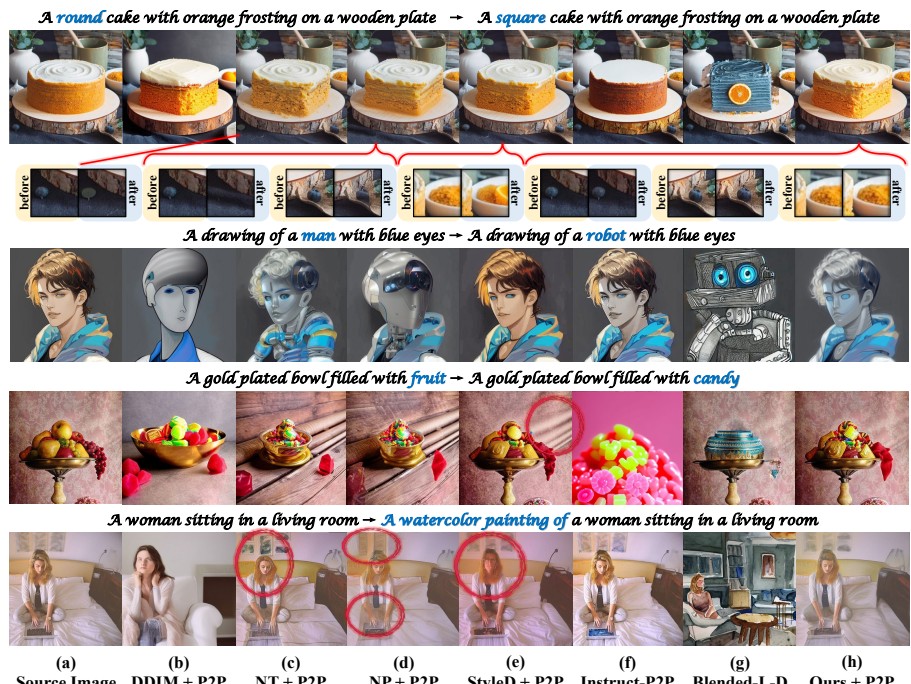

Figure 4: **Visualization results of different inversion and editing techniques.** The source image is shown in col (a). We compare (h) *PnP Inversion* with different inversion techniques added with Prompt-to-Prompt (Hertz et al., 2023): (b) DDIM Inversion (Song et al., 2020), (c) Null-Text Inversion (Mokady et al., 2023), (d) Negative-Prompt Inversion (Miyake et al., 2023), and (e) StyleDiffusion (Li et al., 2023b). We also compare model-based editing results: (f) Instruct-Pix2Pix (Brooks et al., 2023) and (g) Blended Latent Diffusion (Avrahami et al., 2023). The improvements are mostly tangible, and we circle some of the subtle discrepancies w/o *PnP Inversion* in red.

Table 1 shows the structure retention, background preservation, and edit CLIP Similarity of the four inversion methods and the four editing methods. Results show that when added with *PnP Inversion*, all editing methods have a performance improvement on the retention of background and structure while improving or maintaining editability compared with other inversion methods. While DDIM Inversion may yield a high edit CLIP Similarity within the edited mask, the preservation of structure and background falls significantly below the acceptable performance threshold, as depicted in Figure 4. We also give out the inference time of Negative-Prompt Inversion (NP), Edit Friendly Inversion (EF), AIDI (Pan et al., 2023), EDICT, Null-Text Inversion (NT), and Style Diffusion (SD) added with Prompt-to-Prompt in Table 2. *PnP Inversion* achieves better editing results with far less inference time than Null-Text Inversion and StyleDiffusion. Although Negative-Prompt Inversion and Edit Friendly DDPM infer a little faster than *PnP Inversion*, their editing results are much more unacceptable compared to *PnP Inversion* as shown in Table 1 and Table 3.

## 5.3 COMPARISON WITH ESSENTIAL CONTENT PRESERVATION METHODS

We compare *PnP Inversion* with inversion and editing techniques targeted for essential content preservation in Table 3. Null-Text Inversion (NT) (Mokady et al., 2023) added with Prompt-to-Prompt (P2P) (Hertz et al., 2023) provides a baseline for all improvement methods. Specifically, Negative-Prompt Inversion (NP) (Miyake et al., 2023) maintains a guidance scale of 1 to reduce the deviation in editing. Proximal Guidance (Prox) (Han et al., 2023) limits edit changes to a specific area based on editing amplitude. Edit Friendly DDPM (EF) (Huberman-Spiegelglas et al., 2023) changes the DDPM sampling distribution to allow reconstruction of the desired image. EDICT (Wallace et al., 2023) maintains two coupled noise vectors to invert each other for image reconstruction. Although some of these techniques improve the structure and back-

| Method | | CLIP Similariy | |
|---|---|---|---|
| Inverse | Edit | Whole ↑ | Edit ↑ |
| NT | P2P | 24.75 | 21.86 |
| NT | Prox | 22.91↓ | 20.23↓ |
| NP | Prox | 24.28↓ | 21.36↓ |
| EF | P2P | 23.97↓ | 21.03↓ |
| EDICT | P2P | 23.09↓ | 20.32↓ |
| Ours | P2P | 25.02↑ | 22.10↑ |

Table 3: **Compare *PnP Inversion* with background preservation methods.**

ground preservation compared to Null-Text Inversion, clip similarity has decreased for all methods, which indicates a deteriorating editing ability. On the contrary, *PnP Inversion* can lift structure/background preservation and editability simultaneously, as shown in Table 1.

## 5.4 ABLATION STUDY

### 5.4.1 COMPARING *PnP Inversion* AND NULL-TEXT INVERSION

To validate our theoretical analysis, we prove experimentally in Table 4 that *PnP Inversion*'s improvement over Null-Text Inversion (NT) is a three-step process, disentangling the source and target branch, wiping off the force assignment of null-text embedding, and removing the distance gap shown in Figure 2. To disentangle the two branches, we revise Null-Text Inversion to a single-branch version (NT-S) and only assign the learned null-text latent to the source branch. Results show an improvement in CLIP Similarity, revealing the benefit of leaving the target branch unaltered. To wipe off the force assignment, we use the optimization strategy of Null-Text Inversion, and instead of replacing null-text embedding, we directly add the difference to the source latent. The result is shown as Null-Latent Inversion (NL). To show the influence of the distance gap, we add scaled distance ($scale * d_t^{rec}$) to the source latent. Results show that with the distance gap increase, the structure and background preservation decline, while the edit fidelity fluctuates. Moreover, the Null-Latent Inversion's performance is between added distance with a scale of 0.4 and 0.8, which implies the average optimization distance gap of Null-Text inversion is between 0.4 and 0.8.

| Metrics | Structure | | Background Preservation | | | | CLIP Similarity | |
|---|---|---|---|---|---|---|---|---|
| **Method** | **Distance**$_{\times 10^3}$ ↓ | **PSNR** ↑ | **LPIPS**$_{\times 10^3}$ ↓ | **MSE**$_{\times 10^4}$ ↓ | **SSIM**$_{\times 10^2}$ ↑ | | **Whole** ↑ | **Edited** ↑ |
| **NT**[†] | 13.44 | 27.03 | 60.67 | 35.86 | 84.11 | | 24.75 | 21.86 |
| **NT-S**[†] | 14.25↑ | 26.39↓ | 66.62↑ | 40.09↑ | 83.52↓ | | 25.01↑ | 22.11↑ |
| **Scale** | **Distance**$_{\times 10^3}$ ↓ | **PSNR** ↑ | **LPIPS**$_{\times 10^3}$ ↓ | **MSE**$_{\times 10^4}$ ↓ | **SSIM**$_{\times 10^2}$ ↑ | | **Whole** ↑ | **Edited** ↑ |
| **0.4** | 13.55 | 26.65 | 58.79 | 36.98 | 84.29 | | 25.02 | 22.10 |
| **NL**[†] | 12.05↓ | 27.03↑ | 55.83↓ | 33.94↓ | 84.55↑ | | 25.02↑ | 22.09↑ |
| **0.8** | 11.90 | 27.14 | 54.76 | 33.35 | 84.66 | | **25.08** | **22.12** |
| **1** | **11.65**↓ | **27.22**↑ | **54.55**↑ | **32.86**↓ | **84.76**↑ | | 25.02↑ | 22.10↑ |

[†] averaged results on A800 and RTX3090 since different environment leads to slightly different performance

Table 4: **Ablation study of comparing Null-Text Inversion and *PnP Inversion*.**

### 5.4.2 INFLUENCE OF ADDING DIFFERENCE TO TARGET LATENT

In Algorithm 1, we only add the distance of the source prompt to the source latent. To show the rationality of this operation and the disentanglement of the source and target branch, we compare the performance of adding source/reconstruction distance $d_{t-1}^{rec}$ to the target latent and adding target distance $d_{t-1}^{tgt}$ to the target latent in Table 5, where the operation on source branch is leaved unchanged. Specifically, $d_{t-1}^{tgt}$ is computed by changing $C^{src}$ to $C^{tgt}$ in Algorithm 1 line 10. Adding source distance to the target latent leads to a decline in both structure/background preservation and clip similarity. Although adding target distance to the target latent leads to better structure/background preservation, the clip similarity (edit fidelity) sharply decreases.

| Metrics | Structure | | Background Preservation | | | | CLIP Similariy | |
|---|---|---|---|---|---|---|---|---|
| **Add** | **Distance**$_{\times 10^3}$ ↓ | **PSNR** ↑ | **LPIPS**$_{\times 10^3}$ ↓ | **MSE**$_{\times 10^4}$ ↓ | **SSIM**$_{\times 10^2}$ ↑ | | **Whole** ↑ | **Edit** ↑ |
| $d_{t-1}^{rec}$ | 19.30 | 26.15 | 63.70 | 46.45 | 83.67 | | 24.93 | 21.88 |
| $d_{t-1}^{tgt}$ | **9.86** | **27.66** | **50.17** | **33.28** | **85.13** | | 23.00 | 20.27 |
| 0 | 11.65 | 27.22 | 54.55 | 32.86 | 84.76 | | **25.02** | **22.10** |

Table 5: **Results of adding the difference to the target latent.**

## 6 CONCLUSION

This paper introduces *PnP Inversion*, a simple yet effective technique for inverting diffusion models. By disentangling the source and target branches in diffusion-based editing, *PnP Inversion* separates the responsibility for essential content preservation and edit fidelity, thus achieving superior performance in both aspects. To address the lack of standardized performance criteria for inversion and editing techniques, we develop *PIE-Bench* comprising 700 images in natural and artificial scenes featuring ten distinct editing types. Evaluation metrics demonstrate that *PnP Inversion* outperforms eight editing methods across five inversion techniques in terms of both edit quality and inference speed. Limitations and future work can be found in supplementary files.

**Acknowledgment.** This work was supported in part by the Innovation and Technology Fund under Grant No. MRP/022/20X, and in part by Research Matching Grant (CSE-7-2022) - RMG01.

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

**Reproducibility Statement.** To ensure the reproducibility and completeness of this paper, we include the Appendix with 7 sections. Appendix A provides details of related works, offering additional information to complement the main text. Appendix B introduces the construction of *PIE-Bench* in detail and provides examples in the benchmark. Appendix C illustrates the evaluation metrics we use in our experiments. Appendix D contains the details of our implementation. Appendix E contains quantitative results on the reconstruction ability of different inversion methods, a full comparison with essential content preservation methods, a comparison with model-based editing, ablation of step and interval, influence of inverse and forward guidance scale, and results of different editing types. Appendix F provides more qualitative results compared with different inversion-based editing, essential content preservation methods, and model-based editing. Lastly, we include limitations and future works in Section G.

## A    RELATED WORK

Diffusion models excel in multiple generation tasks (Liu et al., 2022b;a; 2023c;b; Ju et al., 2023; Ma et al., 2023a;b; Chen et al., 2023a;b; Lu et al., 2023). As mentioned in the main paper, diffusion-based image editing involves two primary concerns: (1) edit fidelity and (2) essential content preservation. Most diffusion-based editing methods take both aspects into consideration and perform editing using a two branches strategy, *i.e.*, a source diffusion branch to maintain the source image's essential content and a target diffusion branch to insert editing instruction (shown in Figure 5). Accordingly, we have a comprehensive review of prior methodologies concerning both two aspects.

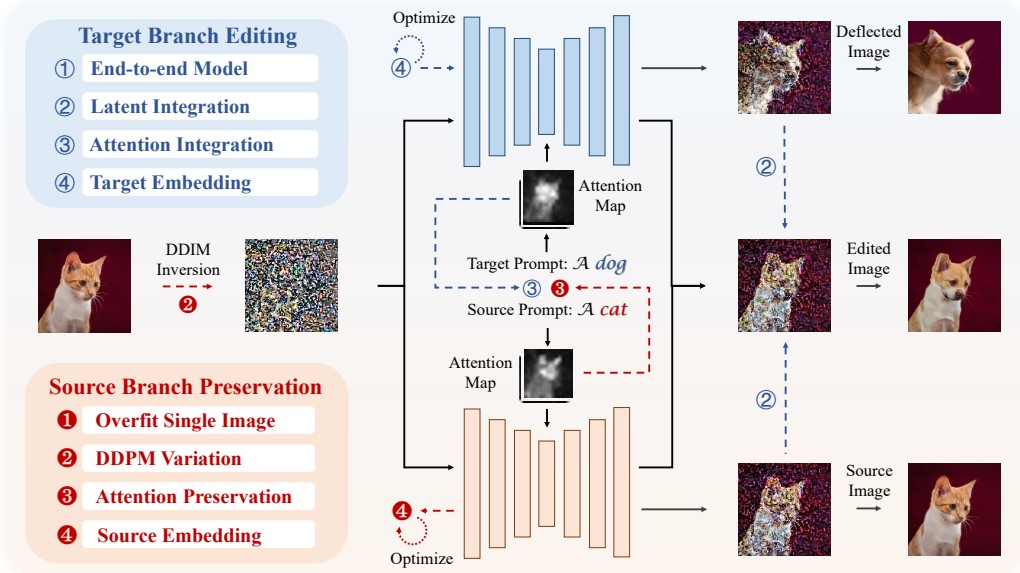

Figure 5: **Diffusion-based editing pipeline** showing how edit fidelity and essential content preservation are achieved in different methods. Detailed introduction is in supplementary files.

**Edit Fidelity.** Diffusion models naturally possess hierarchical features (*e.g.*, noisy latent of each step, different resolution UNet features), enabling different editing strategies. Previous methods perform editing roughly through four ways: ① end-to-end editing model (with only one editing branch), ② latent integration, ③ attention integration, and ④ target embedding.

Specifically, ① trains end-to-end diffusion models for image editing, which is limited by insufficient/noisy training data or indirect training strategies (Brooks et al., 2023; Kim et al., 2022; Nichol et al., 2022; Mirzaei et al., 2023). The shared objective of ②-④ is to map both the source image and the target editing instruction to the diffusion space, then inject the target branch's features into the source diffusion space. ② inserts editing instruction in the level of the noisy diffusion latent (Meng et al., 2022; Avrahami et al., 2022; 2023; Couairon et al., 2023; Zhang et al., 2023b; Joseph et al., 2023; Elarabawy et al., 2022).The features of the source and the target branch are merged through

mask stitching (Meng et al., 2022; Avrahami et al., 2022; 2023; Couairon et al., 2023; Joseph et al., 2023) or weighted addition (Zhang et al., 2023b; Shi et al., 2023). However, using mask stitching for feature insertion may lead to abrupt editing boundaries, and using weighted addition makes it difficult to make refined modifications. ③ tries to solve these two problems by fusing in a more refined feature space, the attention map that connects the text and image. Prompt-to-prompt (Hertz et al., 2023) directly replaces the cross-attention map to perform editing through text. Proximal guidance (Han et al., 2023), Zero-shot (Parmar et al., 2023), MasaCtrl (Cao et al., 2023), Plug-and-Play (Tumanyan et al., 2023), RIVAL (Zhang et al., 2023a), and DragonDiffusion (Mou et al., 2023) further extend the use of both cross-attention and self-attention map to achieve better editing results or explore more applications. ④ first aggregates editing information into an embedding, then uses this embedding to perform editing on the source diffusion branch, which may confront long feature extraction times and unstable editing performance (Kawar et al., 2023; Cheng et al., 2023; Wu et al., 2023; Brack et al., 2023; Tsaban & Passos, 2023; Valevski et al., 2022; Dong et al., 2023).

| Method | Source Branch | | | | Target Branch | | | | Method | Source Branch | | | | Target Branch | | | |
|---|---|---|---|---|---|---|---|---|---|---|---|---|---|---|---|---|---|
| | ❶ | ❷ | ❸ | ❹ | ① | ② | ③ | ④ | | ❶ | ❷ | ❸ | ❹ | ① | ② | ③ | ④ |
| Mokady et al. (2023) | | | | ✓ | | | | | Cao et al. (2023) | | ✓ | | | ✓ | | | |
| Mou et al. (2023) | | ✓ | | | ✓ | ✓ | | | Brooks et al. (2023) | ✓ | | | | ✓ | | | |
| Kim et al. (2022) | ✓ | ✓ | | | ✓ | | | | Nichol et al. (2022) | ✓ | | | | ✓ | | | |
| Mirzaei et al. (2023) | | | | | ✓ | | | | Meng et al. (2022) | | | | | | ✓ | | |
| Avrahami et al. (2023) | | | | | | ✓ | | | Avrahami et al. (2022) | | | | | | ✓ | | |
| Couairon et al. (2023) | | | | | | ✓ | | | Zhang et al. (2023b) | ✓ | | | | | ✓ | | |
| Shi et al. (2023) | ✓ | | | | | ✓ | | | Hertz et al. (2023) | | | ✓ | | | | ✓ | |
| Han et al. (2023) | | ✓ | | | | | ✓ | | Parmar et al. (2023) | | | ✓ | | | | ✓ | |
| Tumanyan et al. (2023) | | | ✓ | | | | ✓ | | Zhang et al. (2023a) | ✓ | ✓ | | | | | ✓ | |
| Kawar et al. (2023) | ✓ | | | | | | | ✓ | Cheng et al. (2023) | | | ✓ | | | | | ✓ |
| Wu et al. (2023) | | | ✓ | | | | | ✓ | Brack et al. (2023) | | | | | | | | ✓ |
| Tsaban & Passos (2023) | | | | | | | | ✓ | Valevski et al. (2022) | ✓ | | | | | | | ✓ |
| Dong et al. (2023) | | | ✓ | | | | | ✓ | Miyake et al. (2023) | | ✓ | | | | | | |
| Huberman-Spiegelglas et al. (2023) | ✓ | | | | | | | | Qi et al. (2023) | | | ✓ | | | | ✓ | |
| Li et al. (2023b) | | | ✓ | | | ✓ | | | Gal et al. (2022) | | | | ✓ | | | | ✓ |
| Fei et al. (2023) | | | ✓ | | | | | | Huang et al. (2023) | | | | ✓ | | | | |
| Wallace et al. (2023) | | ✓ | | | | | | | Joseph et al. (2023) | | | | | | ✓ | | |
| Geng et al. (2023) | | | ✓ | | | | | | | | | | | | | | |

Table 6: **Strategies for enhancing editing fidelity and preserving essential content in previous diffusion-based editing methods.**

**Essential Content Preservation.** While the methods mentioned enable basic image editing, preserving the essential content, particularly on images devoid of inherent diffusion space, remains challenging. Previous methods tried to solve this problem through ❶ overfit the editing image, ❷ DDPM/DDIM inversion variation, ❸ attention preservation, and ❹ source embedding.

Specifically, ❶ overfits the source image to avoid significant image variation (Kawar et al., 2023; Shi et al., 2023). ❷ makes variations on the DDPM/DDIM sampling process to adapt the editing. Negative-prompt Inversion (Miyake et al., 2023) set the classifier-free guidance scale to 1 to reduce the deviation caused by DDIM inversion, which weakens the text's controllability. Edit Friendly Noise (Huberman-Spiegelglas et al., 2023) imprints the source image more strongly onto the noise space to ensure better reconstruction. However, this reduces the modification space due to the decrease in noise. EDICT (Wallace et al., 2023) maintains two coupled noise vectors to reach mathematically exact inversion, but leading to a decrease of edit fidelity. ❸ devises ways of utilizing both cross-attention and self-attention map with better balance of semantic editing results and original image structure (Mou et al., 2023; Cheng et al., 2023; Cao et al., 2023; Parmar et al., 2023; Tumanyan et al., 2023; Hertz et al., 2023; Qi et al., 2023). ❹ absorbs source image to an embedding and use this embedding to reconstruct the essential content of the source image (Mokady et al., 2023; Dong et al., 2023; Li et al., 2023b; Gal et al., 2022; Fei et al., 2023; Huang et al., 2023). Specifically, Null-text inversion (Huang et al., 2023) optimizes a Null embedding to capture the difference between the reconstructed image and the source image. Subsequently, this difference is

steply reintroduced in both source and target branch during the editing procedure. However, null-text inversion necessitates prolonged optimization times per image, lacks the assurance of achieving flawless optimization, and disturbs the diffusion model distribution. Prompt Tuning Inversion (Dong et al., 2023) and StyleDiffusion (Li et al., 2023b) optimize text embedding and cross-attention value to capture the difference instead of null-text, thus facing the same issue with Null-text Inversion.

More refined categorization is presented in Table 6. To summarize, existing background preservation methods suffer from unstable and time-consuming optimization processes, as well as the persisting issue of error propagation inversion. Moreover, the absence of a disentanglement for the source and target branches is unfavorable for achieving optimal performance in both edit fidelity and essential content preservation. Instead, a simple yet effective *PnP Inversion* is capable of achieving superior results with virtually negligible computational cost and negligible inversion error without optimization by branch disentanglement, aiding in accurately editing the real images while preserving the structural information.

## B  BENCHMARK CONSTRUCTION

Although diffusion-based editing has been widely explored in recent years, people mainly evaluate the performance of different editing methods with subjective and incomprehensive visualization results. Previously, PnP (Tumanyan et al., 2023) provides a benchmark of 55 images with editing prompts. Instruct-Pix2Pix (Brooks et al., 2023) builds a dataset with randomly selected 451,990 images, editing prompts written by ChatGPT, and pseudo editing results of Null-Text Inversion (Mokady et al., 2023) and Prompt2Prompt (Hertz et al., 2023). However, without manual labels and fine-grained classification, these datasets cannot support comprehensive metrics evaluation.

To systematically validate our proposed method as a plug-and-play strategy for editing models and compare our method with existing inversion methods, as well as compensate for the absence of standardized performance criteria for inversion and editing techniques, we construct a benchmark dataset, named *PIE-Bench* (**P**rompt-based **I**mage **E**diting **Bench**mark).

*PIE-Bench* comprises 700 images in natural and artificial scenes (*e.g.*, paintings) featuring ten distinct editing types as shown in Figure 3: (0) random editing written by volunteers (140 images), (1) change object (80 images), (2) add object (80 images), (3) delete object (80 images), (4) change object content (40 images), (5) change object pose (40 images), (6) change object color (40 images), (7) change object material (40 images), (8) change background (80 images), and (9) change image style (80 images). In each editing type of 1-9, images are evenly distributed among natural and artificial scenes. Within each scene, images are evenly distributed among four categories: animal, human, indoor environment, and outdoor environment. Each image in *PIE-Bench* includes five annotations: a source image prompt, a target image prompt, an editing instruction, edit subjects describing the main editing body, and the editing mask. For editing type 0, we invited some volunteers to write the source image prompt, target image prompt, and editing instructions based on their editing expectations. For the other editing types, we employ BLIP-2 (Li et al., 2023a) to generate the source image prompt and use GPT4 (OpenAI, 2023) to craft the target image prompt and editing instructions tailored to each editing type. Then, we manually rectify any inaccuracies in the automatically generated captions, target prompt, and edit instructions. Subsequently, 2 data annotators and 1 data auditor annotate the main editing body as well as the editing mask (indicating the anticipated editing region) in an image. Notably, the editing mask annotation is crucial in accurate metrics computations as we expect the editing to only occur within the designated area.

## C  EVALUAION METRICS

To illustrate the effectiveness and efficiency of our proposed *PnP Inversion*, we use eight metrics covering four aspects: structure distance, background preservation, edit prompt-image consistency and inference time.

**Structure Distance:** We follow Tumanyan et al. (2022) to leverage self-similarity of deep spatial features extracted from DINO-ViT as a structure representation and use cosine similarity between image features as structure distance. The structure distance can capture structure while ignoring

appearance information. Thus, it is well-suited for our proposed benchmark and diffusion-based editing methods since we do not expect a huge structural change.

**Background Preservation:** We calculate standard PSNR, LPIPS (Zhang et al., 2018), MSE, and SSIM (Wang et al., 2004) in the area outside of the manual-annotated masks of *PIE-Bench* to demonstrate the background preservation ability of different inversion and editing techniques.

**Edit Text-image Consistency:** The CLIP (Radford et al., 2021) Similarity (CLIPSIM (Wu et al., 2021)) evaluates text-image consistency between the edited images and corresponding target editing text prompts. CLIP Similarity projects text and images to the same shared space and evaluates the similarity of their embeddings. We calculate CLIP Similarity both on the whole image and in the editing mask (black out everything outside the mask) to demonstrate the performance of editing, as well as reflecting the editability. These two metrics are called Whole Image Clip Similarity and Edit Region Clip Similarity, respectively.

**Inference Time:** We test inference time per image of different inversion techniques and Prompt-to-Prompt (Hertz et al., 2023) on one NVIDIA A800 80G to evaluate efficiency. Results are averaged over 20 random runs.

## D    IMPLEMENTATION DETAILS

We perform the inference of different editing and inversion methods in the same setting unless specifically clarified, *i.e.*, on RTX3090 following their open-source code with a base model of Stabe Diffusion v1.4 in 50 steps, with an inverse guidance scale of 1 and a forward guidance scale of 0. Different images may have different hyper-parameters in different editing models, and we keep the recommended hyper-parameter for each editing method in all images for fair comparison. Details can be found in the provided code.

## E    QUANTITATIVE RESULTS

### E.1    RECONSTRUCTION ABILITY OF DIFFERENT INVERSION METHODS

To further show the reconstruction ability of different inversion methods, we evaluate the reconstruction results of DDIM Inversion, Null-Text Inversion, Negative-Prompt Inversion, StyleDiffusion, and *PnP Inversion* by giving source prompt as model input. We provide results of Structure Distance and Background preservation to show the ability to correct $z_t^{''}$ back to $z_t^*$. As shown in Table 7, *PnP Inversion* is better than all these inversion methods on all metrics.

| Inverse | Structure Distance$_{\times 10^3}$ ↓ | PSNR ↑ | LPIPS$_{\times 10^3}$ ↓ | MSE$_{\times 10^4}$ ↓ | SSIM$_{\times 10^2}$ ↑ |
|---------|-------------------------------------|--------|-------------------------|----------------------|------------------------|
| **DDIM** | 70.23 | 17.76 | 210.84 | 224.43 | 70.96 |
| **NT†** | 3.30 | 30.17 | 33.39 | 18.86 | 86.84 |
| **NP** | 8.47 | 27.73 | 57.04 | 30.05 | 84.59 |
| **StyleD** | 4.35 | 28.88 | 39.45 | 22.63 | 86.07 |
| **Ours** | **2.95** | **30.57** | **31.41** | **17.60** | **87.20** |

[†] averaged results on A800 and TRX3090 since different environment leads to slightly different performance

Table 7: **Reconstruction results of different inversion techniques**

### E.2    COMPARISON WITH ESSENTIAL CONTENT PRESERVATION METHODS

We provide the full table comparing *PnP Inversion* with inversion and editing techniques targeted for background preservation in Table 8. As explained in the main paper, although some of these techniques improve the structure and background preservation compared to Null-Text Inversion, clip similarity has decreased for Proximal Guidance (Prox), Negative-Prompt Inversion (NP), Edit Friendly Inversion, and EDICT, which indicates a deteriorating editing ability. However, *PnP Inversion* can lift structure/background preservation and editability simultaneously, which shows the effectiveness of *PnP Inversion* in image editing.

| Method | Structure | Background Preservation | | | | CLIP Similariy | |
|---|---|---|---|---|---|---|---|
| **Inverse** | **Editing** $\mid$ **Distance**$_{\times 10^3}$ $\downarrow$ | **PSNR** $\uparrow$ | **LPIPS**$_{\times 10^3}$ $\downarrow$ | **MSE**$_{\times 10^4}$ $\downarrow$ | **SSIM**$_{\times 10^2}$ $\uparrow$ | **Whole** $\uparrow$ | **Edit** $\uparrow$ |
| **NT** | P2P | 13.44 | 27.03 | 60.67 | 35.86 | 84.11 | 24.75 | 21.86 |
| **NT** | Prox | **3.51**$\downarrow$ | **30.21**$\uparrow$ | **32.97**$\downarrow$ | **18.47**$\downarrow$ | **87.01**$\uparrow$ | 22.91$\downarrow$ | 20.23$\downarrow$ |
| **NP** | Prox | 7.44$\downarrow$ | 28.67$\uparrow$ | 41.98$\downarrow$ | 24.25$\downarrow$ | 85.98$\uparrow$ | 24.28$\downarrow$ | 21.36$\downarrow$ |
| **EF** | P2P | 18.05$\uparrow$ | 24.55$\downarrow$ | 91.88$\uparrow$ | 94.58$\uparrow$ | 81.57$\downarrow$ | 23.97$\downarrow$ | 21.03$\downarrow$ |
| **EDICT** | P2P | 4.61$\downarrow$ | 29.79$\uparrow$ | 37.03$\downarrow$ | 20.37$\downarrow$ | 86.55$\uparrow$ | 23.09$\downarrow$ | 20.32$\downarrow$ |
| **EDICT** | / | 13.28$\downarrow$ | 26.76$\downarrow$ | 65.51$\uparrow$ | 38.14$\uparrow$ | 83.72$\downarrow$ | 24.46$\downarrow$ | 21.56$\downarrow$ |
| **AIDI** | P2P | 12.19$\downarrow$ | 26.96$\uparrow$ | 57.92$\downarrow$ | 39.82$\downarrow$ | 84.17$\uparrow$ | 24.96$\uparrow$ | 22.01$\uparrow$ |
| **Ours** | P2P | 11.65$\downarrow$ | 27.22$\uparrow$ | 54.55$\downarrow$ | 32.86$\downarrow$ | 84.76$\uparrow$ | **25.02**$\uparrow$ | **22.10**$\uparrow$ |
| **Ours+AIDI** | P2P | 11.54$\downarrow$ | 27.26$\uparrow$ | 54.54$\downarrow$ | 32.78$\downarrow$ | 84.69$\uparrow$ | **25.02**$\uparrow$ | 22.09$\uparrow$ |

Table 8: **Full table of comparing *PnP Inversion* with background preservation methods.** Null-Text Inversion (NT) (Mokady et al., 2023) added with Prompt-to-Prompt (P2P) (Hertz et al., 2023) provides a baseline for all improvement methods. Specifically, Negative-Prompt Inversion (NP) (Miyake et al., 2023) maintains a guidance scale of 1 to reduce the deviation in editing. Proximal Guidance (Prox) (Han et al., 2023) limits edit changes to a specific area based on editing amplitude. Edit Friendly DDPM (EF) (Huberman-Spiegelglas et al., 2023) changes the DDPM sampling distribution to allow reconstruction of the desired image. EDICT (Wallace et al., 2023) maintains two coupled noise vectors to invert each other for image reconstruction. AIDI (Pan et al., 2023) employs an iterative procedure to find a fixed-point solution for the ideal diffusion latent.

Accelerated Iterative Diffusion Inversion (AIDI) Pan et al. (2023) is also an essential content preservation approach that employs an iterative procedure to find a fixed-point solution for the ideal diffusion latent, as depicted by the dashed circle in Figure 2. AIDI is distinct from and complementary to our approach; while AIDI concentrates on refining DDIM Inversion, our methodology is aimed at the DDIM Forward correction. We have conducted experiments to both compare and combine our method with AIDI. The table above illustrates that our method outperforms AIDI in terms of structural distance, background preservation, and editability. Additionally, the integration of AIDI into our method results in further enhancements to performance. This improvement substantiates the orthogonality and potential synergistic relationship between our approach and AIDI.

### E.3 COMPARISON WITH MODEL-BASED EDITING

We also compare three model-based editing methods, InstructPix2Pix (Brooks et al., 2023), Instruct-Diffusion (Geng et al., 2023), and Blended Latent Diffusion (Avrahami et al., 2023) in Table 9. *PnP Inversion* added with Prompt-to-Prompt shows a better structure and background preservation as well as better CLIP similarity than the two end-to-end editing models InstructPix2Pix and Instruct-Diffusion. Blended Diffusion uses an explicit mask and only performs editing in the mask. We directly use ground-truth mask in *PIE-Bench* as input, which leads to better background preservation and CLIP similarity score. However, the forced editing makes the editing part incompatible with the background, as shown in Figure 4, and thus having a much larger Structure Distance compared to *PnP Inversion* added with Prompt-to-Prompt.

| Metrics | Structure | Background Preservation | | | | CLIP Similariy | |
|---|---|---|---|---|---|---|---|
| **Method** | **Distance**$_{\times 10^3}$ $\downarrow$ | **PSNR** $\uparrow$ | **LPIPS**$_{\times 10^3}$ $\downarrow$ | **MSE**$_{\times 10^4}$ $\downarrow$ | **SSIM**$_{\times 10^2}$ $\uparrow$ | **Whole** $\uparrow$ | **Edit** $\uparrow$ |
| **InstructPix2Pix** | 57.91 | 20.82 | 158.63 | 227.78 | 76.26 | 23.61 | 21.64 |
| **InstructDiffusion** | 75.44 | 20.28 | 155.66 | 349.66 | 75.53 | 23.26 | 21.34 |
| **Blended Diffusion** | 81.42 | **29.13** | **36.61** | **19.16** | **86.96** | **25.72** | **23.56** |
| **Ours+P2P** | **11.65** | 27.22 | 54.55 | 32.86 | 84.76 | 25.02 | 22.10 |

Table 9: **Comparison of model-based editing results.**

| Guidance Scale | | Structure | Background Preservation | | | | CLIP Similariy | |
|---|---|---|---|---|---|---|---|---|
| Inverse | Forward | Distance$_{\times 10^3}$ ↓ | PSNR ↑ | LPIPS$_{\times 10^3}$ ↓ | MSE$_{\times 10^4}$ ↓ | SSIM$_{\times 10^2}$ ↑ | Whole ↑ | Edit ↑ |
| 0 | 1 | 6.23 | 28.96 | 40.71 | 22.83 | 86.29 | 23.19 | 20.49 |
| 0 | 2.5 | 8.03 | 28.27 | 44.59 | 25.90 | 85.89 | 23.89 | 21.09 |
| 0 | 5 | 10.74 | 27.49 | 50.69 | 30.55 | 85.23 | 24.64 | 21.75 |
| 0 | 7.5 | 13.38 | 26.86 | 56.66 | 35.29 | 84.56 | 25.04 | 22.08 |
| 1 | 1 | **4.77** | **29.70** | **36.32** | **19.84** | **86.61** | 22.97 | 20.28 |
| 1 | 2.5 | 5.74 | 29.19 | 39.57 | 21.81 | 86.35 | 23.87 | 21.01 |
| 1 | 5 | 8.76 | 28.04 | 47.59 | 27.45 | 85.52 | 24.65 | 21.76 |
| 1 | 7.5 | 11.65 | 27.22 | 54.54 | 32.86 | 84.76 | 25.02 | 22.10 |
| 2.5 | 1 | 20.17 | 25.87 | 67.73 | 49.33 | 83.44 | 21.28 | 19.02 |
| 2.5 | 2.5 | 8.90 | 28.73 | 43.37 | 25.77 | 85.91 | 23.50 | 20.80 |
| 2.5 | 5 | 9.01 | 28.27 | 47.05 | 27.77 | 85.55 | 24.62 | 21.70 |
| 2.5 | 7.5 | 11.36 | 27.35 | 54.43 | 33.40 | 84.76 | **25.09** | **22.15** |
| 5 | 1 | 70.65 | 19.85 | 155.46 | 203.41 | 75.92 | 16.38 | 15.84 |
| 5 | 2.5 | 45.16 | 22.63 | 111.55 | 123.84 | 79.59 | 19.95 | 18.11 |
| 5 | 5 | 28.85 | 25.18 | 79.92 | 70.91 | 82.35 | 22.60 | 20.09 |
| 5 | 7.5 | 23.11 | 25.88 | 71.19 | 54.33 | 83.08 | 24.02 | 21.17 |
| 7.5 | 1 | 97.45 | 17.62 | 203.49 | 309.26 | 71.92 | 14.38 | 14.79 |
| 7.5 | 2.5 | 74.55 | 19.43 | 165.31 | 224.15 | 74.84 | 17.19 | 16.34 |
| 7.5 | 5 | 56.22 | 21.53 | 131.04 | 154.97 | 77.57 | 19.82 | 18.07 |
| 7.5 | 7.5 | 47.06 | 22.76 | 113.34 | 120.94 | 79.05 | 21.66 | 19.42 |

Table 10: **Ablation on the influence of guidance scale**

### E.4 INFLUENCE OF GUIDANCE SCALE

In our experiments, we observed a significant impact of the guidance scale on the inversion and forward processes of DDIM, consequently affecting the editing results. Previous studies have typically employed an inversion guidance scale of 1/0, coupled with an empirical forward guidance scale of 7.5. However, no systematic experimental evidence determines the optimal combination of guidance scales for achieving the best editing performance, and elucidates how deviations in these guidance scales affect both reconstruction and editing. Hence, in this section, we present a comprehensive analysis of systematic results addressing this matter based on Table 10 and find that, in fact, an inverse guidance scale of 2.5 and a forward guidance scale of 7.5 reaches the best balance of essential content preservation and edit fidelity.

When keeping the inverse guidance scale constant, we observed that as the forward guidance scales increased gradually, there was an initial improvement in background preservation, followed by a decline. The inflection point was approximately at the inverse guidance scale being equal to the forward guidance scale (*e.g.*, inverse with guidance scale 2.5, forward with guidance scale 2.5). In contrast, the CLIP similarity showed a consistently increasing trend.

Figure 6 enables a clear observation of a noticeable trade-off between essential content preservation and edit fidelity. The abscissa represents the sorted essential content preservation metrics, while the ordinate corresponds to the respective CLIP Similarity, aiming to illustrate the contrasting balance between these two

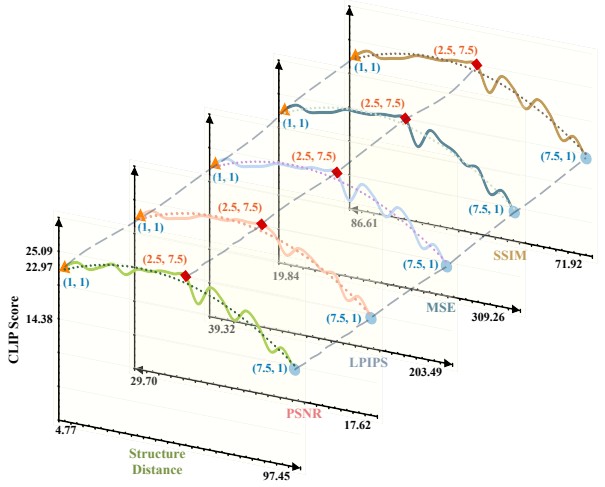

Figure 6: **The impact of different inverse and forward guidance scales on evaluation metrics.**

major categories of metrics. Results show that an inverse guidance scale of 2.5 and a forward guidance scale of 7.5 show the best balance of editing and preservation. The fundamental reason for this trade-off is that present editing methods lack the ability to accurately differentiate between regions that require modification and those that do not. This leads to an inherent conflict where successful and precise edits lead to substantial alterations of the source image while contradictory to the objective of essential content preservation. This experimental observation emphasizes a distinct optimal range for the guidance scale on evaluation metrics. A carefully selected guidance scale can improve the alignment of the inverse and forward processes, and thus improve the editing performance.

### E.5 Ablation of Step and Interval

To illustrate *PnP Inversion*'s performance in different diffusion steps and different add back intervals, we further provide results of *PnP Inversion* added with Prompt-to-Prompt with step numbers of 20, 50, 100, and 500 in Table 11, and with an interval of 1, 2, 5, 10, 24, 49 in Table 12. Results in Table 11 show that *PnP Inversion* is robust to different diffusion steps. Fewer steps will lead to a relatively better preservation of structure and background, bigger steps will have a better clip similarity since target text embedding brings more influence in the inference process. Table 12 shows the results of performing *PnP Inversion* in interval steps, which leads to an update delay. Results show that with the interval increase, performance would become closer to DDIM Inversion with a larger structure/background distance. When the update is performed step-by-step, that is, when the interval is 1, *PnP Inversion* performs best in terms of the overall metrics.

| Metrics | Structure | Background Preservation | | | | CLIP Similarity | |
|---|---|---|---|---|---|---|---|
| **Steps** | **Distance**$_{\times 10^3}$ ↓ | **PSNR** ↑ | **LPIPS**$_{\times 10^3}$ ↓ | **MSE**$_{\times 10^4}$ ↓ | **SSIM**$_{\times 10^2}$ ↑ | **Whole** ↑ | **Edited** ↑ |
| **20** | **10.60** | **27.49** | **51.80** | **30.62** | **84.94** | 24.73 | 21.75 |
| **50** | 11.65 | 27.22 | 54.55 | 32.86 | 84.76 | 25.02 | 22.10 |
| **100** | 12.22 | 27.01 | 56.00 | 34.49 | 84.53 | 25.18 | 22.28 |
| **500** | 13.00 | 26.82 | 57.44 | 35.76 | 84.39 | **25.30** | **22.42** |

Table 11: **Ablation of different inference steps.**

| Metrics | Structure | Background Preservation | | | | CLIP Similarity | |
|---|---|---|---|---|---|---|---|
| **Interval** | **Distance**$_{\times 10^3}$ ↓ | **PSNR** ↑ | **LPIPS**$_{\times 10^3}$ ↓ | **MSE**$_{\times 10^4}$ ↓ | **SSIM**$_{\times 10^2}$ ↑ | **Whole** ↑ | **Edited** ↑ |
| **1** | **11.65** | **27.22** | 54.55 | **32.86** | **84.76** | 25.02 | 22.10 |
| **2** | 11.83 | 27.15 | **54.53** | 33.32 | 84.67 | **25.06** | 22.11 |
| **5** | 13.06 | 26.86 | 57.30 | 35.32 | 84.43 | 25.05 | 22.11 |
| **10** | 16.18 | 26.03 | 66.37 | 42.24 | 83.60 | 24.98 | 22.13 |
| **24** | 24.08 | 24.20 | 77.73 | 66.99 | 82.41 | 24.91 | 22.05 |
| **49** | 47.05 | 21.21 | 128.64 | 126.34 | 78.19 | 24.81 | **22.22** |

Table 12: **Ablation of performing *PnP Inversion* in interval steps.**

### E.6 Results of Different Editing Types

We provide the performance in each editing type of *PnP Inversion* added to Prompt-to-Prompt (Hertz et al., 2023) and MasaCtrl (Cao et al., 2023) in Table 13 and Table 14. The results vary across different editing types and different editing methods. For both Prompt-to-Prompt and MasaCtrl, type 0 performs quite closely in line with its performance across all categories, as it involves random volunteer-selected images and editorial instructions.

In the editing types that Prompt2Prompt struggles with, such as adding objects (type 2), deleting objects (type 3), and modifying object pose (type 5), the model shows minimal changes, resulting in a relatively better evaluation result in essential content preservation metrics. Anomaly, type 5 of Prompt-to-Prompt shows the highest Clip Similarity on editing object poses. We infer the reason lies in the insensitivity of the CLIP model on the object pose and leads to a similarity in source and target prompt features. And since the source prompt is written by Blip2, which has a high CLIP

similarity to the source image, the images with minor alterations in type 5 tend to have a better CLIP Similarity to the source prompt. Type 8 (change background) and 9 (change style) have a bad Structure Distance because the areas that need modification are relatively large. For type 9, the whole image is required for editing. Thus, we do not report background preservation metric and Whole Image CLIP Similarity, which is the same as Edit Region Clip Similarity.

MasaCtrl demonstrates proficiency in modifying object poses, leading to relatively better results in the respective category 5. Moreover, we find that MasaCtrl is also good at adding and deleting objects, which requires significant image modifications and structure changes. We attribute this reason to the Mutual Self-Attention mechanism of MasaCtrl, which facilitates the model's ability to extract texture information from the source branch and structural information from the target branch. Therefore, it is more friendly to modifications sensitive to structural information changes.

| Metrics | Structure | Background Preservation | | | | CLIP Similarity | |
|---|---|---|---|---|---|---|---|
| Type | Distance$_{\times 10^3}$ ↓ | PSNR ↑ | LPIPS$_{\times 10^3}$ ↓ | MSE$_{\times 10^4}$ ↓ | SSIM$_{\times 10^2}$ ↑ | Whole ↑ | Edited ↑ |
| 0 | 11.73 | 28.58 | 50.07 | 30.18 | 85.87 | 25.09 | 22.66 |
| 1 | 12.60 | 26.74 | 55.76 | 29.75 | 84.34 | 24.48 | 20.11 |
| 2 | 10.09 | 27.55 | 53.68 | 30.71 | 86.72 | 25.12 | 23.20 |
| 3 | **9.65** | 23.97 | 78.78 | 53.06 | 78.37 | 23.76 | 17.56 |
| 4 | 12.59 | 28.12 | 51.97 | 23.13 | 85.85 | 24.85 | 22.58 |
| 5 | 10.08 | 26.57 | 60.23 | 33.42 | 82.98 | **26.24** | 22.38 |
| 6 | 10.53 | 26.40 | 55.34 | 35.91 | 83.54 | 25.55 | 20.95 |
| 7 | 11.21 | **30.39** | 40.38 | **17.47** | **89.14** | 25.81 | 23.87 |
| 8 | 12.23 | 27.40 | **39.03** | 30.70 | 87.68 | 24.34 | 22.01 |
| 9 | 14.50 | - | - | - | - | - | **26.04** |

Table 13: **Performance of *PnP Inversion* added with Prompt-to-Prompt on 10 editing types**.

| Metrics | Structure | Background Preservation | | | | CLIP Similarity | |
|---|---|---|---|---|---|---|---|
| Type textbfDistance$_{\times 10^3}$ ↓ | | PSNR ↑ | LPIPS$_{\times 10^3}$ ↓ | MSE$_{\times 10^4}$ ↓ | SSIM$_{\times 10^2}$ ↑ | Whole ↑ | Edited ↑ |
| 0 | 23.48 | 23.76 | 81.72 | 75.25 | 82.67 | 23.66 | 21.27 |
| 1 | 33.26 | 21.03 | 106.87 | 103.16 | 79.25 | 24.27 | 19.59 |
| 2 | **20.08** | 23.26 | 78.73 | 67.14 | 83.61 | 25.04 | 22.73 |
| 3 | 21.55 | 22.44 | 92.66 | 89.60 | 79.16 | **26.38** | 22.31 |
| 4 | 23.60 | 23.51 | 94.72 | 66.43 | 82.08 | 24.52 | 21.55 |
| 5 | 26.53 | **23.79** | 70.81 | **59.32** | **85.89** | 25.44 | 23.44 |
| 6 | 25.79 | 21.65 | 97.36 | 98.51 | 78.64 | 24.62 | 19.61 |
| 7 | 23.63 | 23.49 | **62.20** | 61.98 | 85.27 | 23.39 | 21.55 |
| 8 | 26.87 | 21.05 | 104.79 | 97.34 | 76.11 | 24.16 | 17.67 |
| 9 | 22.53 | - | - | - | - | - | **24.49** |

Table 14: **Performance of *PnP Inversion* added with MasaCtrl on 10 editing types**.

## F  QUALITATIVE RESULTS

Due to the page limit, we do not provide lots of visualization results in the main paper. In this section, we provide a comparison of visualization for further verification of quantitative results.

**Comparison with different inversion-based editing.**   Figure 8 shows the comparison of different inversion methods combined with Prompt-to-Prompt (Hertz et al., 2023). Figure 9, Figure 10, and Figure 11 shows the visualization results of MasaCtrl (Cao et al., 2023), Pix2Pix-Zero (Parmar et al., 2023), and Plug-and-Play (Tumanyan et al., 2023) w/ and w/o *PnP Inversion*.

**Comparison with essential content preservation methods.**   Visualization results of essential content preservation methods are shown in Figure 12, including Proximal Guidance (Han et al., 2023), Edit Friendly DDPM (Huberman-Spiegelglas et al., 2023), and EDICT (Wallace et al., 2023).

**Comparison with model-based editing.** Visualization results of model-based editing are shown in Figure 13, including InstructPix2Pix (Brooks et al., 2023), InstructDiffusion (Geng et al., 2023), and Blended Latent Diffusion (Avrahami et al., 2023).

# G LIMITATIONS AND FUTURE WORKS

Since the performance of *PnP Inversion* is strongly connected to existing diffusion-based editing methods, our method inherits most of their limitations. Although *PnP Inversion* boosts existing editing techniques' performance, it is still unable to bring about fundamental changes in the editing model performance, which is unstable, with a low success rate. In Figure 7, we have chosen specific cases in which Blended Latent Diffusion, along with the ground truth mask, succeeds, whereas all other editing methods fail. This demonstrates the inherent capability of diffusion models to perform corresponding edits. However, existing diffusion-based editing algorithms lack the capability of realization without giving explicit masks. Moreover, although *PnP Inversion* leads to a better performance on average, success is not guaranteed in every case.

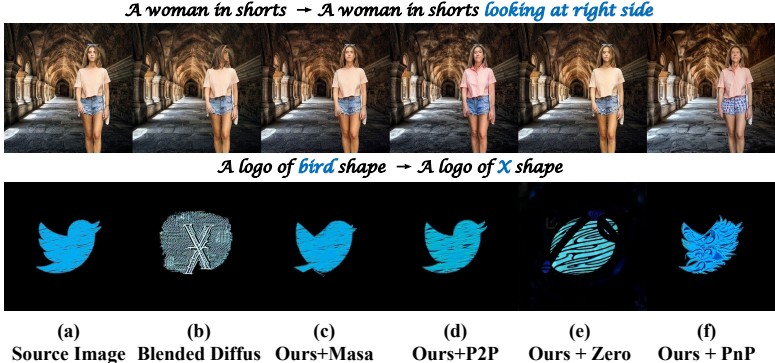

Figure 7: **Visualization results of failure cases in existing diffusion-based editing methods.** (a) source image; (b) Blended Latent Diffusion (Blended Diffus) (Avrahami et al., 2023); (c) *PnP Inversion* added to MasaCtrl (Masa) (Cao et al., 2023); (d) *PnP Inversion* added to Prompt-to-Prompt (P2P) (Hertz et al., 2023); (e) *PnP Inversion* added to Pix2Pix-Zero (Zero) (Parmar et al., 2023); (f)*PnP Inversion* added to Plug-and-Play (PnP) (Tumanyan et al., 2023). The source and target prompt are shown at the top of each row.

*PnP Inversion* may also lead to ethical issues that are worthy of consideration. The data used in the training of diffusion models unavoidably contain personally identifiable information, social biases, and violent content, which will also influence the editing results of our model. *PnP Inversion* can be misused or modified to produce contradictory results and lead to potential negative societal impacts (*e.g.*, arbitrary modification on private photo). We believe these issues should be considered, and we need to design and engineer AI capabilities to fulfill their intended functions while possessing the ability to detect and avoid unintended consequences and unintended behavior.

We hope that this work can motivate future research with a focus on diffusion-based editing for higher essential content preservation and edit fidelity. Specifically, future directions include but are not limited to (1) an extension to video editing, (2) editing models with higher success rates and more editable scenes, and (3) a more comprehensive metric evaluation system to evaluate the effectiveness of editing.

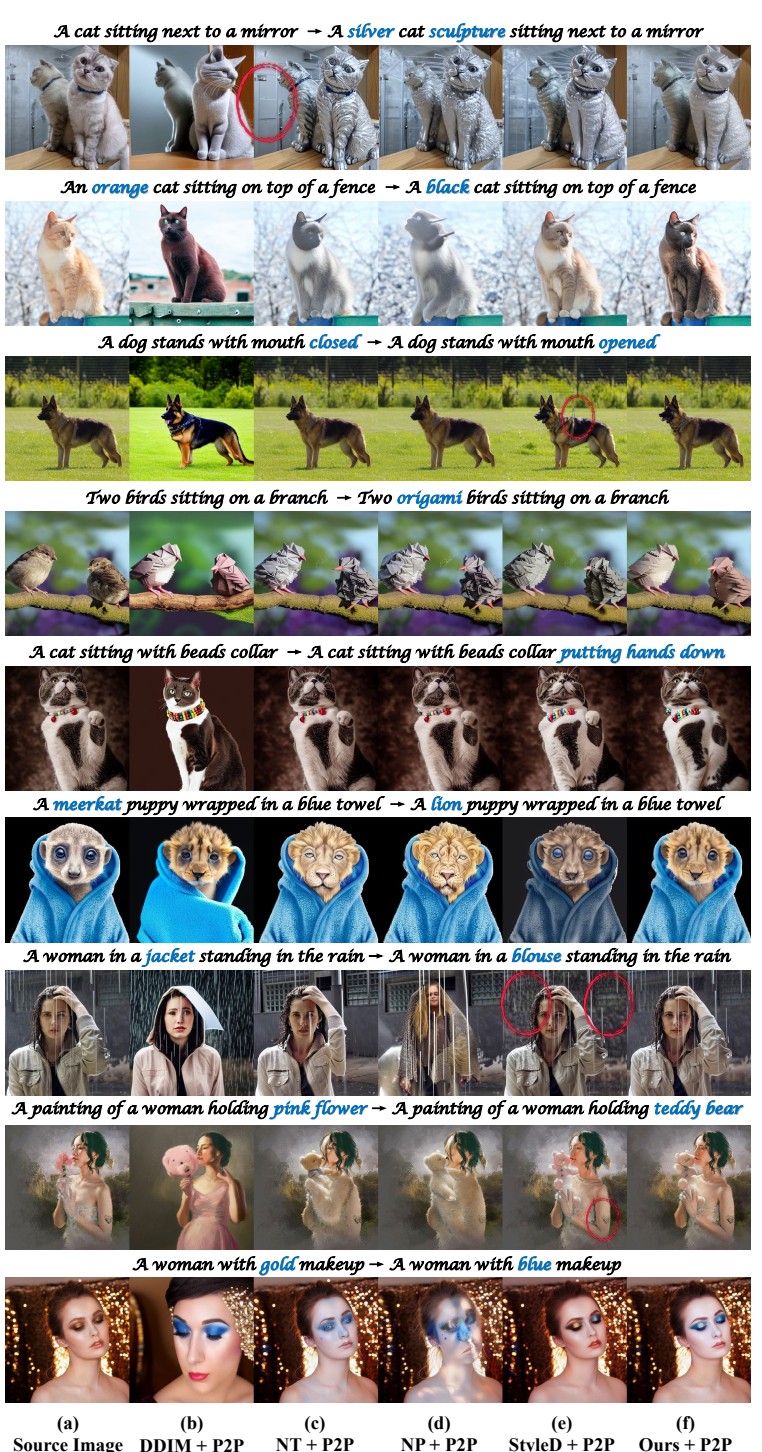

Figure 8: **Visualization results of different inversion methods combined with Prompt-to-Prompt (P2P) (Hertz et al., 2023).** The source image is shown in col (a). We compare (f) *PnP Inversion* with different inversion techniques: (b) DDIM Inversion (DDIM) (Song et al., 2020), (c) Null-Text Inversion (NT) (Mokady et al., 2023), (d) Negative-Prompt Inversion (NP) (Miyake et al., 2023), and (e) StyleDiffusion (StyleD) (Li et al., 2023b). The source and target prompt are shown at the top of each row. The improvements are mostly tangible, and we circle some of the subtle discrepancies w/o *PnP Inversion* in red.

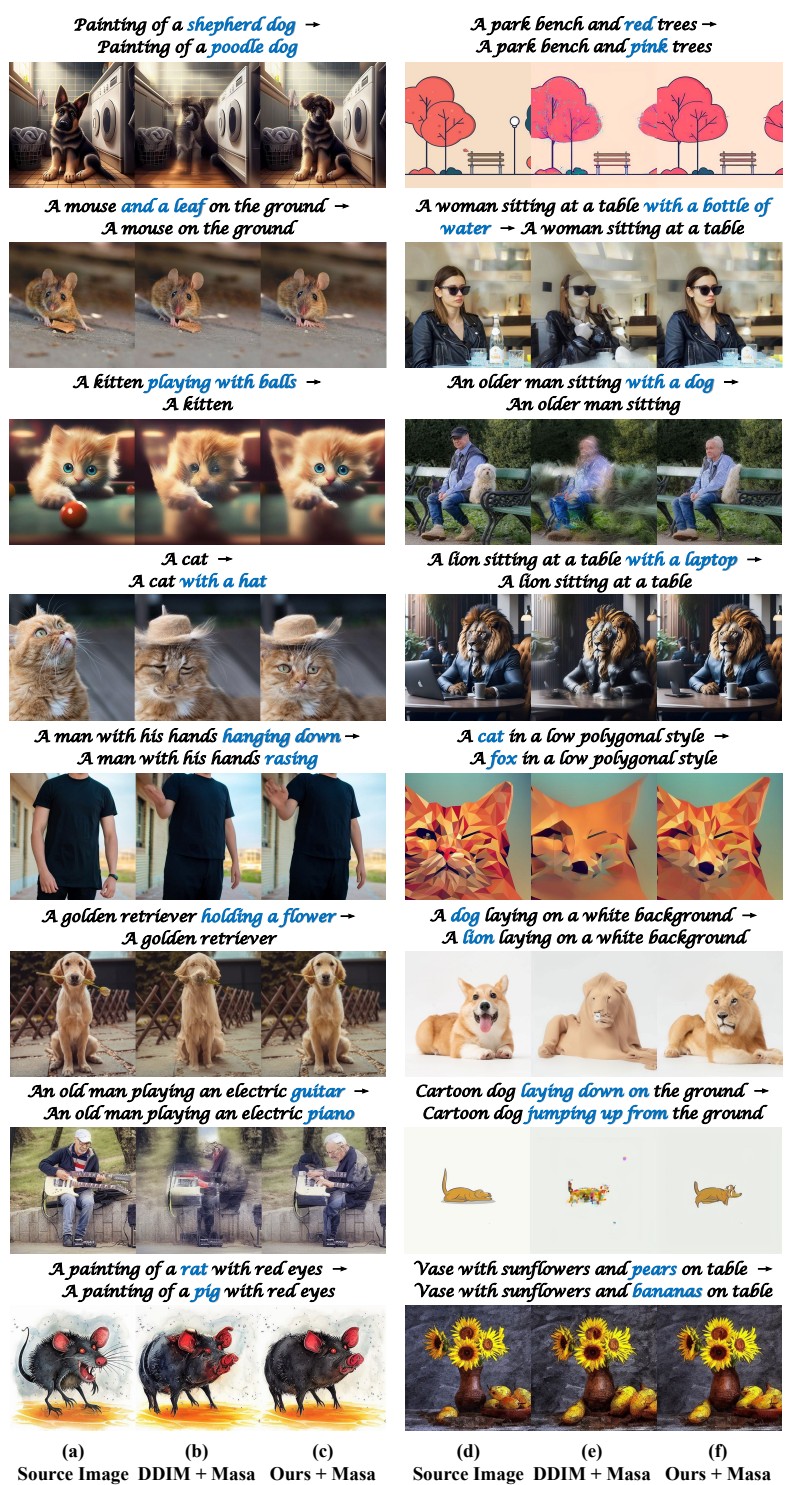

Figure 9: **Visualization results of MasaCtrl (Masa) (Cao et al., 2023) w/ and w/o *PnP Inversion*.** The source image is shown in col (a) and (d). The col (b) and (e) show results w/o *PnP Inversion*. The col (c) and (f) show results w/ *PnP Inversion*. The source and target prompt are shown at the top of each row.

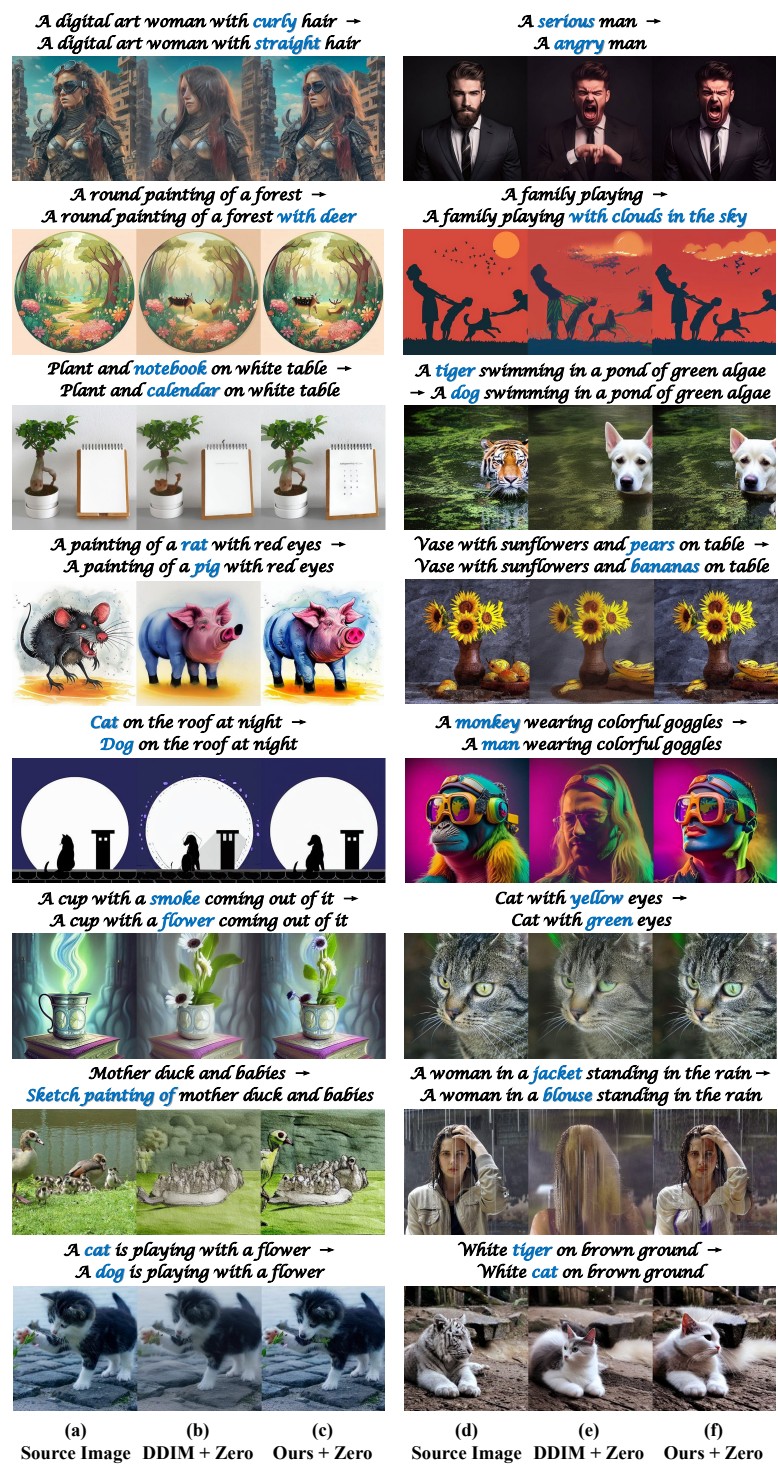

Figure 10: **Visualization results of Pix2Pix-Zero (Zero) (Parmar et al., 2023) w/ and w/o *PnP Inversion*.** The source image is shown in col (a) and (d). The col (b) and (e) show results w/o *PnP Inversion*. The col (c) and (f) show results w/ *PnP Inversion*. The source and target prompt are shown at the top of each row.

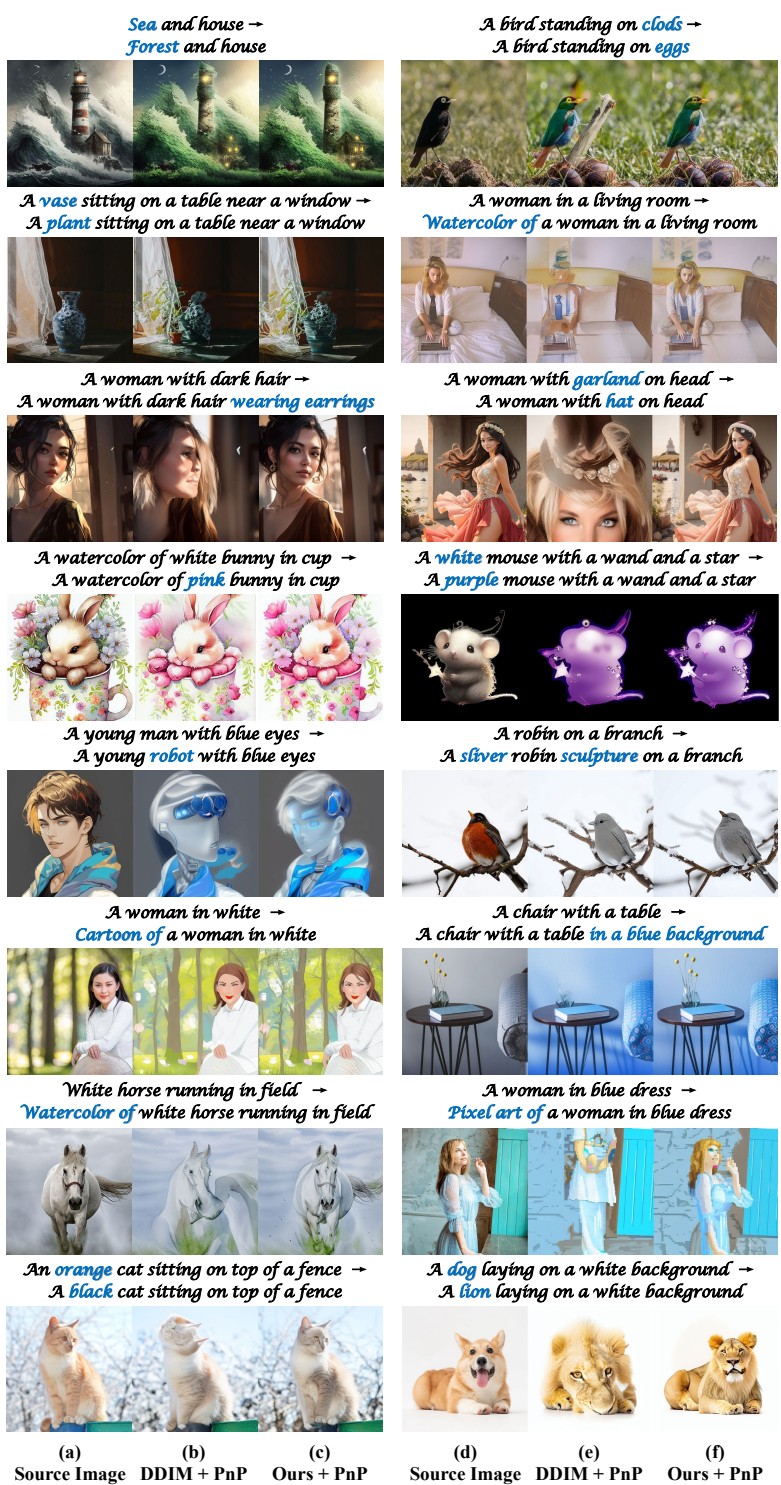

Figure 11: **Visualization results of Plug-and-Play (PnP) (Tumanyan et al., 2023) w/ and w/o**
*PnP Inversion*. The source image is shown in col (a) and (d). The col (b) and (e) show results w/o
*PnP Inversion*. The col (c) and (f) show results w/ *PnP Inversion*. The source and target prompt are
shown at the top of each row.

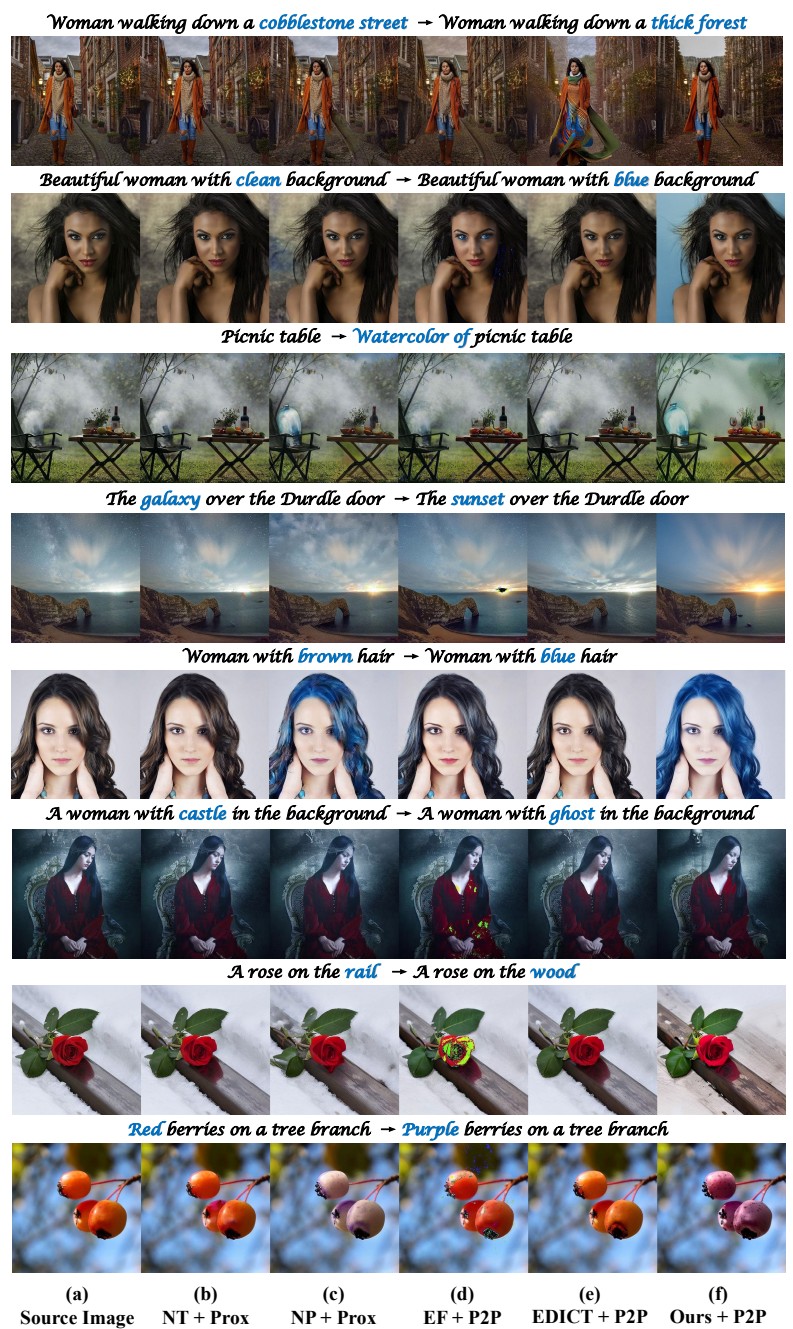

Figure 12: **Visualization results of different essential content preservation methods.** The source image is shown in col (a). We compare (f) *PnP Inversion* added to Prompt-to-Prompt (P2P) (Hertz et al., 2023) with different combined methods: (b) Null-Text Inversion (NT) (Mokady et al., 2023) added to Proximal Guidance (Prox) (Han et al., 2023), (c) Negative-Prompt Inversion (NP) (Miyake et al., 2023) added to Proximal Guidance, (d) Edit Friendly DDPM (EF) (Huberman-Spiegelglas et al., 2023) added to Prompt-to-Prompt, and (e) EDICT (Wallace et al., 2023) added to Prompt-to-Prompt. The source and target prompt are shown at the top of each row.

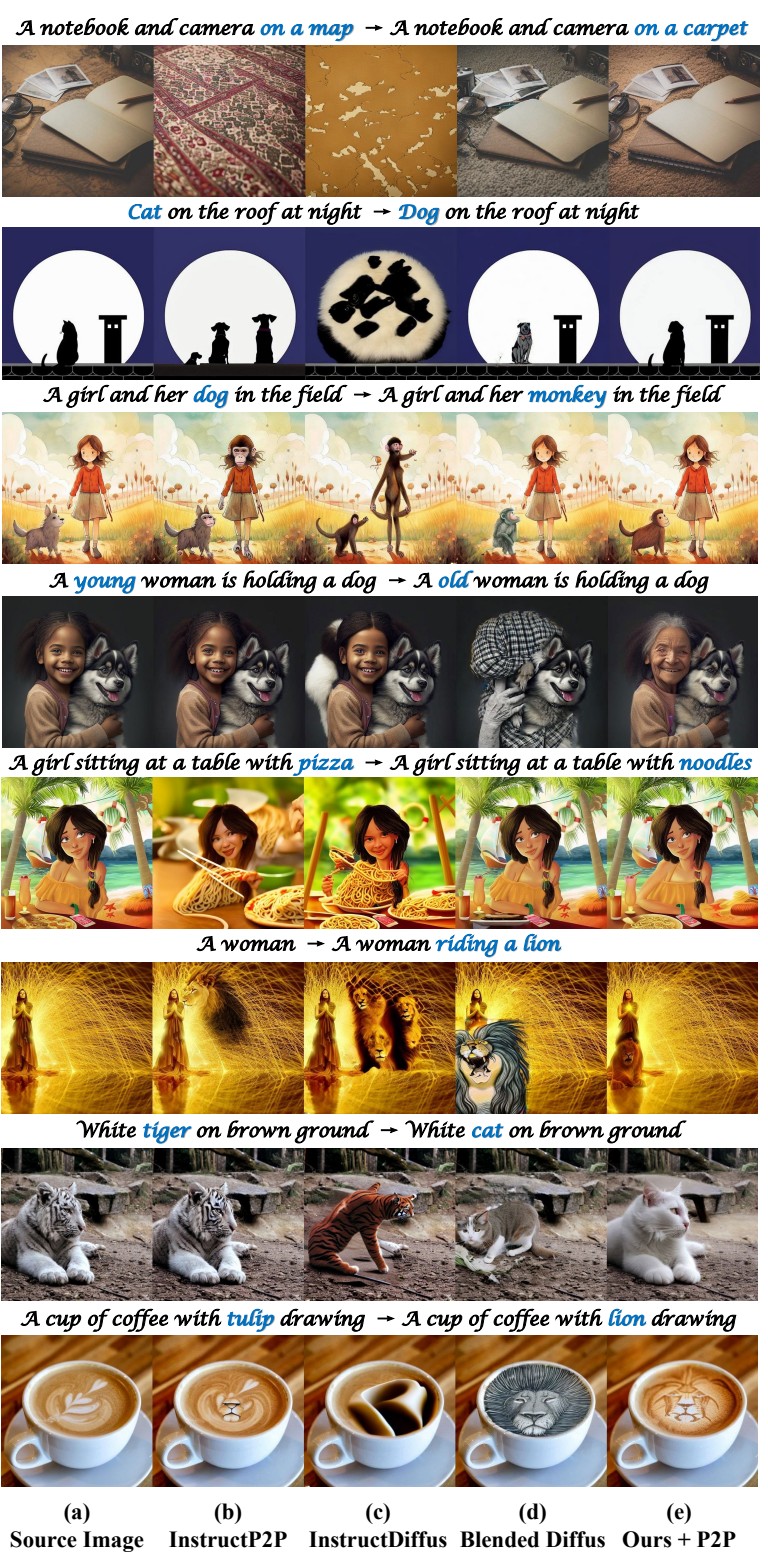

Figure 13: **Visualization results of different essential content preservation methods.** The source image is shown in col (a). We compare (f) *PnP Inversion* added to Prompt-to-Prompt (P2P) (Hertz et al., 2023) with different model-based editing methods: (b) InstructPix2Pix (InstructP2P) (Brooks et al., 2023), (c) InstructDiffusion (InstructDiffus) (Geng et al., 2023), and (d) Blended Latent Diffusion (Blended Diffus) (Avrahami et al., 2023). The source and target prompt are shown at the top of each row.

