# OpenReview forum: "PnP Inversion: Boosting Diffusion-based Editing with 3 Lines of Code"
_ICLR.cc/2024/Conference — ICLR 2024 poster_

### Official Review · Reviewer_jyta · 2023-10-30

**Soundness:** 3 good
**Presentation:** 3 good
**Contribution:** 3 good
**Rating:** 6
**Confidence:** 4

**Summary:**

The $\textbf{Direct Inversion}$ technique separates the source and target diffusion branches, enhancing content preservation and edit fidelity. It surpasses previous methods and substantially accelerates editing, as evidenced by the PIE-Bench benchmark.

**Strengths:**

This paper proposes a novel technique called Direct Inversion, which tackles the problem of balancing content preservation and edit fidelity in previous works. Motivation is clear and presentation is in a roughly good shape overall.

1. Direct Inversion is neat and can be used as a plug-and-play into popular optimizaiton-based diffusion editing methods to enhance the performances.

2. The paper provides a comprehensive and well-structured review of existing literature. Analysis of each method makes the motivation strong and presentation clear.

3. Experimental results are sound and analysis is rigorous.

4. Authors also provide a editing benchmark, called PIE-Bench, which is believed to benefit future works.

**Weaknesses:**

See questions.

**Questions:**

1. In the column PnP of Fig.1, PnP doesn't do a good job in preserving content, sometimes texture and shape is hallucinated. Using direct inversion can correct them. But why not background color in the third row?

2. Following Fig.2, "...This results in a learned latent with a discernible gap between $z_0^{''}$ and the original $z_0$...". So how does the optimized $z_0^{''}$ deviates from original distribution? It's not quite clear how deviation happens, what does it look like, and why it negatively affects performances. Could authors provide concrete examples?

---

> ### Author Response · Authors · 2023-11-16
> **Response to Reviewer jyta**
>
> We would like to sincerely thank the reviewer for his valuable insights and acknowledgment of our study. Following the feedback, we have revised our manuscript, specifically clarifying previously ambiguous areas. We believe these enhancements significantly contribute to the clarity and impact of our research.
>
> > **Question1: Why the proposed inversion method can not correct PnP’s background.**
> >
>
> **Answer1:**
>
> Since Plug-and-Play (PnP) does not have a background protection mechanism but performs editing globally, it often fails to preserve the background effectively. Although our method can improve essential content preservation, it is highly related to the editing model (The $DDIM$ _ $Forward_{Editing}$ _ $_{Model}$) base. Our proposed method aims to maximize the efficacy of existing editing rather than instituting fundamental changes. As evidenced in Figure 1, our approach successfully amends specific background areas, marked in red, in line with our objectives. The further advancement of the performance of these editing methods is designated as an area for future research, as detailed in Section G of the main paper.
>
> > **Question2: What does the deviation from $z_0$ to $z_0^{''}$ looks like.**
> >
>
> **Answer2:**
>
> The deviation of pure DDIM Inversion and DDIM Forward is shown in Figure 4 (b) and Figure 8 (b). The deviation would completely distort the original image content, making it hard to perform essential content preservation.

---

### Official Review · Reviewer_C1kK · 2023-10-31

**Soundness:** 3 good
**Presentation:** 2 fair
**Contribution:** 3 good
**Rating:** 8
**Confidence:** 2

**Summary:**

This paper proposes a simple but effective method that improves existing diffusion-based image editing methods. The method is very easy to implement, with only three lines of code. However, it helps resolve the discrepancy between source latent and target editing latent for many diffusion-based editing methods. This helps preserve essential content and maintain editing flexibility. The paper also presents a comprehensive image editing benchmark PIE-Bench covering ten editing types.

**Strengths:**

1. This paper presents a thorough investigation of existing diffusion-based prompt editing methods and identifies that previous methods improve essential content preservation through fine-tuning a learned variable to rectify the distance between the source branch latent and the target branch latent and propose a simple rectification method.
2. The proposed rectification method is simple but effective and is suitable for a large amount of diffusion-based editing methods. This paper presents comprehensive experiments to apply their method to different methods and see a universal improvement in both essential content preservation and edit fidelity.
3. This paper presents a comprehensive diffusion-based editing benchmark covering ten editing types and 700 human-reviewed high-quality samples with source prompt, target prompt, source image, and editing region annotation, which is helpful for future study.

**Weaknesses:**

1. The writing in the method part is hard to follow. I would suggest authors use meaningful subscripts to denote source latent, source prompt forward latent, and other patents instead of $z_0, z_t', z_{t}''$.

**Questions:**

1. What is the meaning of step 1, 2 in Figure 2?
2. In the algorithm1, what is the meaning of argument $[C^{src}, C^{tgt}]$ in the DDIM_Forward function call? Is it suggesting that the forward function is called twice, with one of the calls on $C^{src}$ and the other on $C^{tgt}$? If so, why line 9 function call has only one output?
3. I am confused about how the source branch interacts with the target branch. It seems that the $z_{t}^{tgt}$ is only updated using $z_{t+1}^{tgt}, C^{tgt}$ without any source branch information. Could authors clarify line 9 in the algorithm? Is it related to the implementation of $DDIMForward_{Editing_Model}$?
4. What's the number of images for each editing type in the PIE-Bench creation? As far as I understand, most of the editing types are local region editing, e.g., change object, add object, and large region editing is limited to style change only. I wonder if the dataset is mostly local editing images.
5. What is the input text for CLIP similarity evaluation in **Whole** and **Edit**? As far as I understand, the target prompt in the PIE-Bench is a full description of the target image instead of the specification of the local region.
6. The CLIP Similarity for different methods seems very close in Table 1, 4, and 8. Is it possible the CLIP similarity cannot distinguish the editing quality? Could authors include BLIP similarity or human evaluation to make evaluation more comprehensive?

I would increase my rating if the authors could resolve the questions.

---

> ### Author Response · Authors · 2023-11-16
> **Response to Reviewer C1kK (1/3)**
>
> We express our sincere gratitude to the reviewer for the insightful comments and the recognition of our work. We especially appreciate the acknowledgment of our approach's simplicity, effectiveness, and superior performance. In response to the feedback, we have meticulously refined our paper and clarified unclear parts in the revised version. We believe these enhancements significantly contribute to the clarity of our research.
>
> > **Question1: Use meaningful subscripts to denote source latent, source prompt forward latent, and other patents.**
> >
>
> **Answer1:**
>
> We sincerely thank the reviewer for pointing out a better writing approach for articulating and explaining our methods. We have revised the denotations as follows:
>
> - We use $z_{0}^{src}$ to denote the source latent.
> - We use $z_{t}^{I}$ to denote DDIM **i**nversion latent. Specifically, we use $z_{t}^{I}$ to denote the nonexistent ideal DDIM inversion latent, and use $z_{t}^{I_{p}}$ to denote the perturbed DDIM inversion latent.
> - We use $z_{t}^{F}$ to denote DDIM **f**orward latent. Specifically, we use $z_{t}^{F_s}$ to denote the DDIM forward latent with **s**ource prompt without guidance scale, and use $z_{t}^{F_t}$ to denote the DDIM forward latent with **t**arget prompt without guidance scale. To further distinguish the DDIM forward process with guidance scale, we use $z_{t}^{F_{sg}}$ to denote the DDIM forward latent with **s**ource prompt with guidance scale, and use $z_{t}^{F_{tg}}$ to denote the DDIM forward latent with **t**arget prompt with guidance scale.
>
> We also revised some other denotations:
>
> - We use $C^{src}$ to denote source prompt embedding (or embedding of null for some editing methods), and use $C^{tgt}$ to denote target prompt embedding.
> - We use $d_{t}^{rec}$ to denote the difference between the perturbed DDIM inversion latent $z_{t}^{I_{p}}$ and the DDIM forward latent of the source prompt $z_{t}^{F_{sg}}$.
>
> > **Question2: The meaning of step 1,2 in Figure2.**
> >
>
> **Answer2:**
>
> The step denotes the order of steps in the editing process.
>
> Specifically, for techniques such as Null-Text Inversion, StyleDiffusion, and Negative-Prompt Inversion, the process involves initially inverting $z_{0}^{src}$ to $z_{1}^{I_p}$ (step 1) followed by inverting it to $z_{2}^{I_p}$ (step 2). Subsequently, editing on $z_{2}^{I_p}$ is executed using a two-branch diffusion model to compute the denoised latent $z_{1}^{F_{sg}}$ and $z_{1}^{F_{tg}}$(step 3), and eventually $z_{0}^{F_{sg}}$ and $z_{0}^{F_{tg}}$(step 4).
>
> For our method, steps 1 and 2 mirror those in Null-Text Inversion, StyleDiffusion, and Negative-Prompt Inversion. Post these steps, editing on $z_{2}^{I_p}$ is conducted using the two-branch diffusion model to determine the denoised latent $z_{1}^{F_{sg}}$ and $z_{1}^{F_{tg}}$(step 3), followed by step 4, which involves adding differences to rectify deviations. Steps 5 and 6 are repetitions of steps 3 and 4. Notably, steps 4 and 6, introduced by our method, are denoted by a solid circle. A comprehensive explanation of these steps is provided in the main paper.

---

> ### Author Response · Authors · 2023-11-16
> **Response to Reviewer C1kK (2/3)**
>
> > **Question3: Better explanation of the algorithm.**
> >
>
> **Answer3:**
>
> We apologize for the lack of clarity and unclear denotations of our proposed algorithm in the historical version. We have revised the algorithm and polished section 4.2 according to your constructive advice:
>
> - **The bracket notations:** The bracket notations denote the batch concatenation for the inputs and outputs involved in DDIM Inversion and DDIM Forward processes instead of calling a function twice. Since existing editing methods simultaneously process the source prompt $C^{src}$ and target prompt $C^{tgt}$, with a batch size of 2, to ensure essential content preservation and editing by integrating information between the source and target batches. In adherence to these established methods, we write the algorithm with an input concatenated in batch dimension. We apologize for not specifying it clearly in the previous version of our manuscript and have added the corresponding clarification in the revised version.
> - **The misalignment of batch size in input and output:** We thank the reviewer for pointing out the misalignment. We have revised our manuscript and ensured the batch size is maintained consistently in input and output.
>
> Additionally, we have updated the algorithm in response to constructive feedback from other reviewers. For detailed information, please refer to the revised version of the paper.
>
> > **Question4: How source branch interacts with the target branch.**
> >
>
> **Answer4:**
>
> The interaction is related to the implementation of $DDIM\underline{}Forward_{Editing\underline{}Model}$. We have provided a detailed review of how previous editing methods perform the source and target branches interaction in the related work. For more information, please refer to Section A in the revised version.
>
> To illustrate this interaction, we utilize the Prompt-to-Prompt method as an example. Prompt-to-Prompt uses the cross-attention map in the source branch to enhance or replace the ones in the target branch. The selection of the cross-attention map is based on the correspondence between the cross-attention map and the text prompt. For instance, in transforming an image of a cat to a dog, the method involves using the source prompt 'a cat' and the target prompt 'a dog'. Here, the cross-attention map associated with 'dog' is exchanged with that of 'cat' in the source branch, thereby infusing the edit with relevant information. Concurrently, other attention maps in the source branch are maintained as is, ensuring the preservation of essential content.
>
> This also explains why an accurate source branch latent would be important for essential content preservation during the editing. This is because the feature (such as attention map) in the source branch will be further used in target branch editing.
>
> > **Question5: The distribution of images for each editing type in the PIE-Bench.**
> >
>
> **Answer5:**
>
> We have clarified the number of images in each editing type in the revised Section B, listed as follows:
>
> 0. random editing written by volunteers (140 images)
>
> 1. change object (80 images)
>
> 2. add object (80 images)
>
> 3. delete object (80 images)
>
> 4. change object content (40 images)
>
> 5. change object pose (40 images)
>
> 6. change object color (40 images)
>
> 7. change object material (40 images)
>
> 8. change background (80 images)
>
> 9. change image style (80 images)
>
> We distribute the editing into several classes: random (type 0, 140 images), object-related (type 1-3, 240 images), object attribute change (type 4-7, 160 images), and large region edit (type 8-9, 160 images). Thus, PIE-Bench covers a wider range of editing categories instead of only local editing.

---

> ### Author Response · Authors · 2023-11-16
> **Response to Reviewer C1kK (3/3)**
>
> > **Question6: The use of CLIP Similarity.**
> >
>
> **Answer6:**
>
> - **Include more metrics to make the evaluation more comprehensive:** Since the editing is expected to only perform in a part of the image and not make huge changes in the image structure. It is true the CLIP score varies modestly among different editing methods. However, we hold the view that CLIP score is still possible to distinguish the editing quality since it is aligned with visualization results, shown in the visualization of the main paper.
> To make the evaluation more comprehensive, we added two human evaluation metrics to rate the editing quality. Due to the limited time, we randomly selected 50 images and let two volunteers rate 1-10 source-editing image pairs on two dimensions: editing image quality and alignment with the target prompt. Both two volunteers rated all 50 images, and we averaged their scores to formulate the table below. The Table shows the same compared methods as in Table 1 of the main paper. For the editing method Prompt-to-Prompt (P2P), we compare four different inversion methods: DDIM Inversion (DDIM), Null-Text Inversion (NT), Negative-Prompt Inversion (NP), and StyleDiffusion. For editing methods MasaCtrl, Pix2Pix-Zero (P2P-Zero), and Plug-and-Play (PnP), we compare with DDIM Inversion (DDIM). Results show that the human evaluation metrics show the same tendency with CLIP Similarity scores.
>
>
>     | Method | Whole Image CLIP Similarity  | Editing Image Quality | Alignment with Target Prompt | Average of Human Evaluation Metrics |
>     | --- | --- | --- | --- | --- |
>     | DDIM+P2P | 25.01 | 1.74 | 5.68 | 3.71 |
>     | NT+P2P | 24.75 | 6.32 | 5.19 | 5.76 |
>     | NP+P2P | 24.61 | 5.19 | 4.65 | 4.92 |
>     | StyleD+P2P | 24.78 | 6.21 | 5.15 | 5.68 |
>     | Ours+P2P | 25.02 | 6.50 | 5.52 | 6.01 |
>     | DDIM+MasaCtrl | 23.96 | 4.54 | 3.88 | 4.21 |
>     | Ours+MasaCtrl | 24.38 | 4.61 | 4.01 | 4.31 |
>     | DDIM+P2P-Zero | 22.80 | 2.86 | 3.01 | 2.94 |
>     | Ours+P2P-Zero | 23.31 | 4.33 | 4.20 | 4.27 |
>     | DDIM+Plug-and-Play | 25.41 | 6.39 | 5.60 | 6.00 |
>     | Ours+Plug-and-Play | 25.41 | 6.56 | 5.61 | 6.09 |
>
>     Table 2. Comparing our methods with other inversion techniques across various editing methods on Whole Image CLIP Similarity and Human Evaluation Metrics.
>
>     We will further evaluate the human evaluation metrics on the whole benchmark with 700 images and report the value in the final version.
>
> - **The input text of CLIP Similarity evaluation:** We directly use the target prompt to calculate both CLIP Similarity in Whole and Edit. It is possible that the masked image is not completely aligned with the target prompt for Edit Region CLIP Similarity. However, since most mask regions in PIE-Bench take up around half of the image size, our visualization results show alignment between the Edit Region CLIP Similarity and masked image. Moreover, the same variation trend of Edit Region CLIP Similarity, Whole Image CLIP Similarity, and Human Evaluation Metrics may also demonstrate the reasonableness of using this metric.
> While acknowledging the logical basis of this metric, we concur with the identified potential risks associated with its use. In response, we have decided to switch it to human evaluation metrics in the final version of our work.

---

> ### Comment · Reviewer_C1kK · 2023-11-18
>
> Thanks to the authors for the clarification. I've increased my rating.
>
> I agree that the target prompts in the PIE bench may not effectively demonstrate the local editing quality when assessing CLIP similarity for certain types of edits. I believe the [Editbenchmark](https://imagen.research.google/editor/) dataset is a better fit for evaluating the local editing quality, as it primarily includes prompts concentrating on local editing.
>
> I'm not asking for additional experiments during the rebuttal period, but I believe these results could be more convincing for the EDIT metric. Additionally, I think it's possible to identify a similar subset within the PIE bench with target prompts focused on local regions. Measuring EDIT metric using this subset might be better.

---

> > ### Author Response · Authors · 2023-11-19
> > **Further Response to Reviewer C1kK**
> >
> > We express our sincere gratitude for the reviewer's prompt and insightful feedback.
> >
> > We agree that EditBenchmark is indeed more apt for evaluating local editing quality. However, EditBenchmark predominantly focuses on image inpainting and necessitates user-provided masks, which diverges from the prompt-based, mask-free editing approach our method employs. This divergence presents challenges in directly applying our method to EditBenchmark. Nonetheless, we acknowledge the value of EditBenchmark's analysis in guiding our search for more appropriate evaluation metrics, particularly in terms of the analysis comparing CLIP metrics with human judgments. We will further intend to delve deeper into these metrics to ascertain if more fitting alternatives are available.
> >
> > Additionally, we agree with the suggestion that it is feasible to identify a subset within the PIE benchmark that concentrates on local regions. In line with this, we aim to introduce an additional annotation for each image pair, specifying the targeted local edit. For instance, to modify the sky from cloudy to sunny, we could incorporate a specific **local eidt prompt**, such as "sunny sky," and assess the CLIP Similarity between the sky region and this targeted **local eidt prompt**. This approach would serve as a novel EDIT metric in our evaluation framework. We will work on refining PIE-Bench and further updating the new metric, as well as analyzing EditBenchmark in our final version.
> >
> > Thanks again for the valuable suggestions and constructive feedback!

---

### Official Review · Reviewer_WzRy · 2023-10-31

**Soundness:** 4 excellent
**Presentation:** 3 good
**Contribution:** 2 fair
**Rating:** 6
**Confidence:** 4

**Summary:**

This paper proposed a method for Diffusion based inversion and editing, where all the intermediate generated zt values are stored and then utilized for sampling new samples based solely on the difference from the zt values generated during the sampling process. This approach is easy to apply to all editing methods. The method demonstrated its applicability to P2P, MasaCtrl, P2P-Zero, and PnP by calculating the difference between the zt values during the editing process (with those methods) and the zt values obtained from the original DDIM inversion. To evaluate this method, the paper introduced PIE-Bench, which used 700 images divided into 10 categories for editing to showcase the preservation of structure, background, and CLIP similarity.

**Strengths:**

This paper provided well-organized evaluation criteria, encompassing 10 different categories that include tasks such as changing or adding and removing objects, altering poses, changing colors, modifying materials, and changing backgrounds. Additionally, editing masks are also provided.

Based on these evaluation criteria, the paper presented a wide array of numerical evaluation results, proving superior performance across all categories. A detailed ablation study was provided, and in the supplementary material, various experimental setups and their results were thoroughly documented, offering valuable insights and enhancing reproducibility.

**Weaknesses:**

First and foremost, I would like to make a strong suggestion to the authors. The content of Figure 3 seems to be largely irrelevant to the content of the paper. While I express my utmost gratitude for the detailed explanation and organization of previous works, Figure 3 does not play a significant role in aiding understanding. Instead, I would prefer if Figure 5 from the supplementary material were included in the main text. Additionally, a more detailed explanation of the benchmarks and evaluation metrics in the main body of the text would be beneficial. This information is considered one of the major contributions of the paper, yet it is not present in the main text.

Secondly, the explanation of the method is unclear. There are no definitions provided for what the brackets [ ] mean in lines 3, 7, 8, and 9 of Algorithm 1, or what o_t represents. There is also a need for an explanation on whether z_t encompasses both src and tgt. If my understanding based on the code is correct, this paper stores all the zt values generated during DDIM inversion, uses them to calculate a small editing direction at each step, and then reflects this in the z^tgt used to generate the actual results. This algorithm feels somewhat similar to the approach used in CycleDiffusion [https://arxiv.org/abs/2210.05559]. A clearer explanation of the algorithm would greatly assist in correcting my understanding.

Thirdly, the benchmark is divided into 10 categories, but scores for each category are not reported. I am particularly interested in the scores for the category involving pose changes. I suspect that most of the proposed methodologies would struggle with changing poses. A discussion and reporting of scores on this matter would be appreciated, at least in the supplementary material.

Minor point: Regarding Figure 4, it is disappointing that the only result shown with our method applied is Ours+P2P. (But I saw additional results in Supple.)

**Questions:**

Please see the weakness part.

Especially I'm wondering about the Algorithm.

**Details Of Ethics Concerns:**

Additionally, I would like to highlight the importance of discussing the ethical implications of the presented work in the paper.

---

> ### Author Response · Authors · 2023-11-16
> **Response to Reviewer WzRy (1/2)**
>
> We sincerely thank the reviewer for your insightful comments and recognition of this work, especially for acknowledging the convincing and comprehensive experimental results as well as the contribution of benchmark construction and well-organized evaluation criteria. We have polished the paper, added the experiments, and made the clarifications in the revised version.
>
> > **Question1: Figure 5 is more relevant to the content of the paper than Figure 3. A more detailed explanation of the benchmarks and evaluation metrics in the main body of the text would be beneficial.**
> >
>
> **Answer1:**
>
> We would like to express our sincere thanks to the reviewer for the constructive feedback on the structure of our paper. We fully agree that Figure 5 is more relevant to the content of the paper in comparison with Figure 3 and have accordingly updated the revised manuscript to reflect this change. But we still want to explain why we put substantial effort into introducing previous works. This is because our key insight is the disentangling of two branches, which is ignored by previous works. Based on this insight, we review previous research and dig into the approaches they employed to preserve essential content and facilitate editing. Then, we analyze how to disentangle the two branches in previous works to combine them with our proposed inversion method better. In the revised version, we have put the main part of the related work and Figure 3 into the supplementary file. Moreover, we added Figure 5 to the main text with more explanation.
>
> However, we apologize that due to the page limit, it is hard for us to put a detailed explanation of evaluation metrics into the main text. In this paper, we do not introduce new evaluation metrics but explore existing metrics suitable for the task and incorporate them into our benchmark. We posit that a comprehensive explanation of these metrics in the supplementary files suffices and should not detract significantly from the reader's comprehension.

---

> ### Author Response · Authors · 2023-11-16
> **Response to Reviewer WzRy (2/2)**
>
> > **Question2: The explanation of the method is unclear.**
> >
>
> **Answer2:**
>
> We apologize for the lack of clarity in the explanation of our proposed algorithm and polish section 4.2 according to your constructive advice:
>
> - **The bracket notations:** The bracket notations denote the batch concatenation for the inputs and outputs involved in DDIM Inversion and DDIM Forward processes. Since existing editing methods simultaneously process the source prompt $C^{src}$ and target prompt $C^{tgt}$, with a batch size of 2, to ensure essential content preservation and editing by integrating information between the source and target batches. In adherence to these established methods, we write the algorithm with an input concatenated in batch dimension. We apologize for not specifying it clearly in the previous version of our manuscript and have added the corresponding clarification in the revised version.
> - **The** $o_t$ : The $o_t$ represents the difference between DDIM inversion latent and DDIM forward latent in each time step $t$ . To clarify the notation, we have revised it to $d_t$ , representing the **d**ifference in time step **t**. Moreover, we add a detailed explanation of the calculation of the difference in the main text.
> - **Whether $z_t$ encompasses both src and tgt**: We apologize for the unclear notation in previous manuscripts. In the revised version, we have explicitly distinguished different branches in different batch channels with [] notation. Specifically, the process of generating DDIM Inversion latent is only performed in the source branch. During editing, the source and target embeddings concatenate in the batch dim to simplify the interaction between the two branches.
> - **The comparison to CycleDiffusion:** CycleDiffusion directly calculates an offset between the latent generated by the source prompt and the target prompt of the current time step t in the DDIM Forward process, then adds the offset to the latent embedding of time step t-1. This is quite similar to the methods used in [3][4][5], which use the difference between the source and target prompts to perform image editing. However, our methods are distinguished from their approaches in several aspects: (1) Our method targets editing methods that involve detailed editing operations such as attention map fusion and latent fusion. These editing methods usually perform more satisfying editing results than [3][4][5] and CycleDiffusion. (2) Our method is a plug-and-play improvement for diffusion inversion targeted for two-branch prompt-based editing methods, instead of an editing method. (3) Our method largely improves the essential content preservation and editability by two-branch disentangling, while [3][4][5] and CycleDiffusion directly use a one-branch approach to perform editing.
> Nevertheless, we are thankful to the reviewer for pointing out the missing citation for CycleDiffusion. We add citations for [6][7] to our related work.
>
> [3] General Image-to-Image Translation with One-Shot Image Guidance
>
> [4] SINE: SINgle Image Editing with Text-to-Image Diffusion Models
>
> [5] SEGA: Instructing Diffusion using Semantic Dimensions
>
> [6] Unifying Diffusion Models' Latent Space, with Applications to CycleDiffusion and Guidance
>
> [7] A Latent Space of Stochastic Diffusion Models for Zero-Shot Image Editing and Guidance
>
> > **Question3: Scores for each category.**
> >
>
> **Answer3:**
>
> In fact, we have already provided the results and analysis of our method added to Prompt-to-Prompt on different editing categories in supplementary file E.6. We agree that a discussion and reporting of scores on various editing types would be helpful for a better understanding of the paper. Thus, we additionally add each category's result of MasaCtrl in supplementary file E.6.
>
> Different editing types perform differently in each category. For instance, Prompt-to-Prompt excels in material alterations, resulting in an overall superior performance in category 7. Conversely, MasaCtrl **demonstrates proficiency in modifying object poses** using the mutual self-attention mechanism, leading to relatively better results in the respective category 5. The detailed results and analysis can be found in the supplementary file.
>
> > **Question4: Only the result of Ours+P2P is shown in Figure 4.**
> >
>
> **Answer4:**
>
> The target of Figure 4 is to provide a comparison of different inversion methods and our inversion method. For a fair comparison, all inversion methods are combined with Prompt-to-Prompt. Additional visualization results of our method combined with different editing techniques can be found in Figure 1 and supplementary files from Figure 8 to Figure 13. Due to the page limit, most visualization results are in the supplementary files.

---

### Official Review · Reviewer_2rho · 2023-11-02

**Soundness:** 3 good
**Presentation:** 3 good
**Contribution:** 3 good
**Rating:** 6
**Confidence:** 4

**Summary:**

The paper presents "direct inversion", a general inversion technique to improve essential content preservation and edit fidelity of diffusion-based image editing methods. An editing benchmark is also proposed for performance evaluation.

**Strengths:**

- The paper is well-written and well-presented with nice figures.
- The results are pleasing to look at and are convincing.
- The proposed method is simple and effective.
- Dataset/evaluation benchmark contribution.
- The experiments are comprehensive. The proposed method is quite general and is evaluated on 8 recent editing methods.

**Weaknesses:**

- The method section 4.2 is not very clear to me, especially the bracket notations in the algorithm box. It would be helpful to explain lines 3, 7-9 in more detail.
- It might be worth adding discussion and comparison of a related but concurrent work [1].
- The name "direct inversion" clashes with another existing work [2], which might cause ambiguous.
- Typo: Algorithm 1, Part I: "Invert" z_0^{src}; sec 4.2, "optimization-based" inversion.
- The paper shows promising empirical results but is still not theoretically motivated.

[1] Pan, Zhihong, et al. "Effective Real Image Editing with Accelerated Iterative Diffusion Inversion." Proceedings of the IEEE/CVF International Conference on Computer Vision. 2023.
[2] Elarabawy, Adham, Harish Kamath, and Samuel Denton. "Direct inversion: Optimization-free text-driven real image editing with diffusion models." arXiv preprint arXiv:2211.07825 (2022).

**Questions:**

please see my questions in weakness section.

---

> ### Author Response · Authors · 2023-11-16
> **Response to Reviewer 2rho (1/2)**
>
> We sincerely thank the reviewer for the insightful comments and recognition of this work, especially for acknowledging that our approach is simple and effective with superior performance. We have polished the paper, added the experiments, and clarified the below points in the revised version.
>
> > **Question1: More explanation of the method section.**
> >
>
> **Answer1:**
>
> We apologize for the lack of clarity in the explanation of our proposed algorithm in the historical versions and polish section 4.2 according to your constructive advice:
>
> - **The bracket notations:** The bracket notations denote the batch concatenation for the inputs and outputs involved in DDIM Inversion and DDIM Forward processes. Since existing editing methods simultaneously process the source prompt $C^{src}$ and target prompt $C^{tgt}$, with a batch size of 2, to ensure essential content preservation and editing by integrating information between the source and target batches. In adherence to these established methods, we write the algorithm with an input concatenated in batch dimension. We apologize for not specifying it clearly in the previous version of our manuscript and have added the corresponding clarification in the revised version.
> - **Explain lines 3, 7-9 in more detail:** We have reformulated the algorithm to enhance its readability, adopting a new style of presentation. The revised algorithm and added explanation can be found in the edited rebuttal revision. Specifically, our method achieves editing performance enhancement in 3 steps: (1) we first compute the perturbed latent in DDIM forward; (2) we then store the difference between the original latent in the inversion process and the perturbed latent in the forward process with input condition $C^{src}$; (3) we finally add the difference to the source/reconstruction branch in existing editing model.
>
> We express our gratitude for the reviewer's suggestions, which have significantly improved the quality and readability of the paper.
>
> > **Question2: It worth adding discussion and comparison of work [1].
> [1] Pan, Zhihong, et al. "Effective Real Image Editing with Accelerated Iterative Diffusion Inversion." Proceedings of the IEEE/CVF International Conference on Computer Vision. 2023.**
> >
>
> **Answer2:**
>
> We regret the initial oversight of not providing discussion and comparison with the AIDI method introduced in the current literature. The method under consideration, known as Accelerated Iterative Diffusion Inversion (AIDI), employs an iterative procedure to find a fixed-point solution for the ideal diffusion latent, as depicted by the dashed circle in Figure 2.
>
> It is pertinent to note that AIDI is distinct from and complementary to our approach; while AIDI concentrates on refining DDIM Inversion, our methodology is aimed at the DDIM Forward correction. We have conducted additional experiments to both compare and combine our method with AIDI. The results have been incorporated into the supplementary material in Section E.2 since AIDI reports proficiency in reconstruction. Due to the absence of an official AIDI implementation, we adapted the Prompt-to-Prompt (P2P) codebase to include AIDI with the iteration parameters set to 5 and 20, which is denoted as AIDI(5) and AIDI(20).
>
> | Method | Structure Distance ↓ | PSNR ↑ | LPIPS ↓ | MSE ↓ | SSIM ↑ | $\mathrm{CLIPSimilarity}_{\mathrm{Whole}}$ ↑ | $\mathrm{CLIP Similarity}_{\mathrm{Edited}}$ ↑ |
> | --- | --- | --- | --- | --- | --- | --- | --- |
> | AIDI(5)+P2P | 12.19 | 26.96 | 57.92 | 39.82 | 84.17 | 24.96 | 22.01 |
> | AIDI(20)+P2P | 12.16 | 27.01 | 56.39 | 36.90 | 84.27 | 24.92 | 22.02 |
> | Ours+P2P | 11.65 | 27.22 | 54.55 | 32.86 | **84.76** | **25.02** | **22.10** |
> | Ours+AIDI(5)+P2P | **11.54** | **27.26** | **54.54** | **32.78** | 84.69 | **25.02** | 22.09 |
>
> Table 1. Results of comparing and combining our method with AIDI.
>
> The table above illustrates that our method outperforms AIDI in terms of structural distance, background preservation, and editability. Additionally, the integration of AIDI into our method results in further enhancements to performance. This improvement substantiates the orthogonality and potential synergistic relationship between our approach and AIDI.
>
> Moreover, given that AIDI necessitates an iterative process to locate the fixed-point solution, it is inherently more time-consuming than our proposed method, even when the iteration parameter is limited to 5. Our method achieves an average inversion time of 16.15 seconds, compared to 21.39 seconds for AIDI. These results have been updated in the main body of the paper.

---

> ### Author Response · Authors · 2023-11-16
> **Response to Reviewer 2rho (2/2)**
>
> > **Question3: The name "direct inversion" clashes with another existing work [2].
> [2] Elarabawy, Adham, Harish Kamath, and Samuel Denton. "Direct inversion: Optimization-free text-driven real image editing with diffusion models." arXiv preprint arXiv:2211.07825 (2022).**
> >
>
> **Answer3:**
>
> We thank the reviewer for highlighting the potential for ambiguity with the name ‘Direct Inversion’ used in existing work. To avoid confusion and maintain our methodology's distinctiveness, we have renamed our approach to 'PnP Inversion.' We have updated our manuscript accordingly to reflect this change.
>
> > **Question4: Typo errors.**
> >
>
> **Answer4:**
>
> We appreciate the reviewer's careful reading of our manuscript and the assistance in identifying typographical errors. We have thoroughly reviewed the document and corrected the typos that were pointed out.
>
> > **Question5: The paper shows promising empirical results but is still not theoretically motivated.**
> >
>
> **Answer5:**
>
> While providing mathematical proof for our method is challenging, we offer some theoretical considerations. For more details, please refer to Sections 4.1 and 4.2 of our paper.
>
> We establish our theoretical foundation on the observation that optimization-based inversion techniques attain satisfactory performance by correcting the perturbations in DDIM forward latent, reverting them to their corresponding DDIM inversion counterparts. Despite their efficacy, optimization-based inversion methods are impeded by several limitations: (1) undesirable latent space distances affecting essential content preservation, (2) misalignment in the generation model’s distribution, and (3) extended processing times.
>
> In response to these challenges, we find a simple and effective solution neglected by previous works. This solution adopts the principles of optimization-based methods—namely, the adjustment of DDIM forward latent—while innovating through two pivotal adjustments: (1) disentangling the source and target branches, and (2) empowering each branch to excel in its designated role: preservation or editing.
>
> Using Prompt-to-Prompt as an illustrative example, we further explain why the solution is theoretically reasonable. The key idea of Prompt-to-Prompt is using the cross-attention map in the source branch to enhance or replace the ones in the target branch. As a result, an accurate source branch latent would be important for essential content preservation during the cross-attention map integration. Put simply, a source branch that fails to reconstruct the original image will invariably find it impossible to conserve pertinent information within it. Therefore, correcting the DDIM forward latent back to the DDIM inversion latent is necessary for achieving better structure and background preservation. Simultaneously, the editing branch, targeted for the incorporation of novel concepts, benefits from remaining unmodified and achieving better editability. This motivation is also proved in our ablation study in Section 5.4.

---

### Author Response · Authors · 2023-11-16
**General Response to Common Questions**

We sincerely thank all the reviewers for your constructive feedback and recognition of this work, especially for acknowledging our work’s strengths: 1) **simple and effective plug-and-play method** (Reviewer 2rho, C1kK, jyta), 2) **convincing and comprehensive experimental results** (Reviewer all reviewers),  3) **good presentation** (Reviewer 2rho, jyta), 4) **evaluation benchmark and criteria contribution** (all reviewers), and 5) **a thorough related work** (Reviewer C1kK, jyta).

We would also like to express our sincere gratitude to the reviewers for their insightful identification of areas where our manuscript could be strengthened. We have carefully considered all the suggestions provided and have incorporated substantial revisions to our manuscript. For ease of review, we have highlighted all amendments to the main submission and the supplementary document in **blue** text. We have summarized some common questions and how we revise our manuscript as follows:

- **The paper presentation and writing** (mentioned by Reviewer 2rho, WzRy, C1kK).
    - We have changed our proposed method’s name from **Direct Inversion** to **PnP Inversion**.
    - We have **polished the writing of Algorithm 1** and **added more explanation of our proposed method** in Section 4.
    - We have **revised the notation** of DDIM Inversion and Forward latent in our manuscript, using meaningful subscripts to denote each notation.
    - We added more detailed PIE-Bench explanations and moved part of the related work section to the supplementary file.
    - We have corrected typographical errors and provided additional explanations in previously unclear sections.
- **The explanation of our proposed method** (mentioned by Reviewer 2rho, WzRy, C1kK)
    - Previous inversion methods all perform on both source and target branches in editing, thus leading to suboptimal results.
    The key insight of our proposed methods is that (1) we found a disentangling of source and target branches in editing would make it easier to (2) empower each branch to excel in its designated role.
    - We have added a more detailed explanation with a clearer algorithm in Section 4.2 to express our insight in our manuscript.
- **More comparison of our method**  (mentioned by Reviewer 2rho, WzRy, C1kK)
    - We have added experiments for **comparing and combining AIDI** with our method in the manuscript.
    - We have added **two human evaluation metrics** to rate the editing quality: editing image quality, and alignment with the target prompt. Due to the limited time, we only present the results on a subset of the PIE-Bench. In the final version, we will add scores on the whole benchmark.
    - We have added more discussion and reporting of scores on various editing types in the supplementary file.
    - We have added citations of papers mentioned by the reviewers in the main text.

---

### Meta-Review · Area_Chair_4o8x · 2023-12-06

**Metareview:**

**Summary**

This paper proposes an approach named "PnP inversion" for diffusion-based image editing, which was initially named "direct inversion" but has been changed in the revision due to possible conflict with an existing method. It is simple to implement ("with 3 lines of code"), and yet achieves good editing performance via disentangling the source and target diffusion branches to improve *essential content preservation* and *edit fidelity*. In order to evaluate image-editing performance, this paper also presents an editing benchmark, named PIE-Bench.

**Strengthes**
- The proposal can be combined with any diffusion-based editing method, is easy to implement, and yet is efficient in improving editing performance.
- This paper provides a detailed summary of existing pieces of work in Section 2.
- A benchmark dataset for prompt-based image editing, called PIE-Bench, is presented, which will be useful in comparative studies.

**Weaknesses**
- Some reviewers raised concern about clarity, especially in Algorithm 1 describing the proposal, which however have been addressed in the revision appropriately.
- This paper shows promising empirical results but theoretical motivation is rather weak, although it should be quite difficult to provide a thorough theoretical analysis.

**Justification For Why Not Higher Score:**

The rating/confidence of the 4 reviewers are 6/4, 6/4, 8/2, 6/4, which made me to refrain from higher recommendation. The context of this paper would be somehow limited to image editing with diffusion models, so that it would only attract attention of researcher in the relevant research field.

**Justification For Why Not Lower Score:**

All the reviewers rated this paper positively, above the acceptance threshold. Researchers in the relevant field will benefit not only from the proposal itself, but also from the summary of existing work presented in this paper, as well as the presented benchmark dataset for prompt-based image editing.

---

### Decision · Program_Chairs · 2024-01-16

Accept (poster)